# VLAS: Vision-Language-Action Model with Speech Instructions for Customized Robot Manipulation

**Wei Zhao**[1]   **Pengxiang Ding**[1,2]   **Min Zhang**[1]   **Zhefei Gong**[1]   **Shuanghao Bai**[3]
**Han Zhao**[1,2]   **Donglin Wang**[1]*
[1]Westlake University   [2]Zhejiang University   [3]Xi'an Jiaotong University

## Abstract

Vision-language-action models (VLAs) have become increasingly popular in robot manipulation for their end-to-end design and remarkable performance. However, existing VLAs rely heavily on vision-language models (VLMs) that only support text-based instructions, neglecting the more natural speech modality for human-robot interaction. Traditional speech integration methods usually involves a separate speech recognition system, which complicates the model and introduces error propagation. Moreover, the transcription procedure would lose non-semantic information in the raw speech, such as voiceprint, which may be crucial for robots to successfully complete customized tasks. To overcome above challenges, we propose VLAS, a novel end-to-end VLA that integrates speech recognition directly into the robot policy model. VLAS allows the robot to understand spoken commands through inner speech-text alignment and produces corresponding actions to fulfill the task. We also present two new datasets, SQA and CSI, to support a three-stage tuning process for speech instructions, which empowers VLAS with the ability of multimodal interaction across text, image, speech, and robot actions. Taking a step further, a voice retrieval-augmented generation (RAG) paradigm is designed to enable our model to effectively handle tasks that require individual-specific knowledge. Our extensive experiments show that VLAS can effectively accomplish robot manipulation tasks with diverse speech commands, offering a seamless and customized interaction experience.

## 1 Introduction

With the advent of large vision-language models (VLMs) and the availability of extensive robotic datasets, vision-language-action models (VLAs) (Brohan et al., 2022; 2023; Kim et al., 2024) have become a promising approach for learning policies in robotic manipulation. These models demonstrate enhanced generalization to novel objects and semantically diverse instructions, as well as a range of emergent capabilities. VLAs, such as RT-2 (Brohan et al., 2023), which are fine-tuned from foundation VLMs like PaLM-E (Driess et al., 2023) using robotic trajectory data, can take human instructions and visual observations as inputs to generate robot actions. However, these models primarily focus on textual and visual modalities, leaving the speech modality largely unexplored.

Imagining a scenario where robots provide daily assistance in home care, it is crucial to acknowledge that individuals may exhibit significant variations in physical abilities and subjective preferences. To improve the user experience, robots need to be more accessible and customizable. Speech serves as an ideal modality for achieving this goal, enabling natural and intuitive communication. Given these practical needs and existing technologies, a key question arises: *How can we integrate vision-language-action models with speech modality to produce a simpler and better end-user experience?*

Based on the above analysis, we propose guiding a robot's behavior through speech rather than text. A typical approach involves leveraging an external automatic speech recognition (ASR) system (Radford et al., 2023; Yu et al., 2023) to transcribe speech into text for downstream tasks.

---

*Corresponding author: `wangdonglin@westlake.edu.cn`

However, this method presents two significant issues: Firstly, such a cascading pipeline leads to a larger and more complex robotic system, potentially expanding computational demands and memory consumption. Secondly, the transcription process may lose auxiliary information beyond semantics, such as identity, emotion, and intonation, which are vital for the robot's comprehension of human intent. Many everyday human instructions are unstructured and can only be accurately understood with the support of above auxiliary information from speech. For instance, as illustrated in Figure 1 (a), when given the task "Please pick up my cup", a traditional VLA with text instructions or a VLA incorporating an ASR system may fail to select the correct cup. Therefore, developing a policy model that utilizes raw speech for voice recognition can greatly improve task execution.

To alleviate these two problems, we present VLAS, an innovative end-to-end policy model that seamlessly integrates speech modality for robot manipulation. Notably, VLAS is capable of directly processing both textual and speech instructions alongside visual observations. VLAS is built upon the widely adopted open-source vision-language model, LLaVA (Liu et al., 2023), and is developed through three distinct training phases. Firstly, we employ an established encoder to process speech for hidden representations. The multi-layer perceptrons (MLPs) are fine-tuned to transform these representations into the unified language space of LLaVA. Secondly, we fine-tune the LLaVA model and above MLPs together with multimodal datasets, including our curated Speech Question Answering (SQA) dataset and Visual Question Answering (VQA) datasets. The resulting model, termed VLAS-Base, can effectively generate responses to both text-image and speech-image instructions. Finally, we further fine-tune VLAS-Base through behavior cloning (Ross et al., 2011) on our curated CSI dataset, which encompasses image observations, speech instructions, and robot manipulation trajectories. The voice retrieval-augmented generation (RAG) is subsequently proposed to enable VLAS to perform personalized operations based on individual-specific knowledge. Experimental results show that the proposed VLAS, following either textual or speech instructions, can achieve performance comparable to traditional VLAs on the CALVIN benchmark. In addition, we created a benchmark consisting of customization tasks, where our VLAS demonstrates improved performance by fully leveraging the auxiliary information in speech.

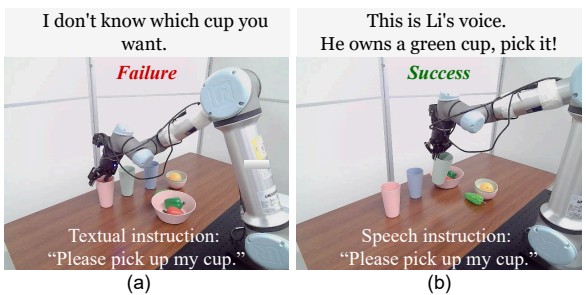

Figure 1: For personalization tasks, (a) previous VLAs with text instructions fail, while (b) our VLAS with speech instructions could successfully address them.

To sum up, the main contributions of this work are listed as follows: 1) We propose VLAS, the first vision-language-action model that integrates speech for robot manipulation without needing external speech recognition systems, enabling more natural communication with robots. 2) A Voice RAG paradigm is designed to enable VLAS to effectively address customized tasks that require individual-specific knowledge. 3) Besides the robot policy model, we introduce VLAS-Base, which extends the widely used vision-language model LLaVA to accept speech instructions. This model is also valuable for other downstream tasks involving speech inputs. We also present two new datasets, SQA and CSI for community further study. The model, data and code will be publicly available at https://github.com/whichwhichgone/VLAS.

## 2 RELATED WORK

**Vision-Language Model** Large language models (LLMs), such as FLAN-PaLM (Chung et al., 2022), InstructGPT (Ouyang et al., 2022), LLaMA (Touvron et al., 2023), and Mamba (Gu & Dao, 2024), trained on web-scale instruction-following datasets, have demonstrated exceptional effectiveness in performing few-shot and zero-shot natural language processing tasks. This approach has also been rapidly adopted in the field of computer vision. Building on these pretrained LLMs, researchers have developed various vision-language models (VLMs), including OpenFlamingo (Awadalla et al., 2023), BLIP-2 (Li et al., 2023), LLaMA-Adapter (Zhang et al., 2024), IDEFICS (Laurençon et al., 2023), Prismatic (Karamcheti et al., 2024), LLaVA (Liu et al., 2023) and Cobra (Zhao et al., 2025),

capable of processing inputs from both text and image modalities simultaneously. Many VLMs tailored for video modalities have also emerged, such as VideoLLaMA (Zhang et al., 2023), VideoLLaMA 2 (Cheng et al., 2024), PiTe (Liu et al., 2024), Video-LLaVA (Lin et al., 2024), and LLaVA-NeXT-Interleave (Li et al., 2024a).

It is worth mentioning that the VLMs discussed in this work refer to models that work in a question-answering format, as opposed to models like CLIP (Radford et al., 2021) and BLIP (Li et al., 2022), which are specifically designed to learn joint representations of linguistic and visual information. Among the prevalent VLMs, LLaVA stands out as a significant milestone due to its full accessibility, reproducibility, and outstanding performance. The key to LLaVA's success lies in its two-stage visual instruction tuning and the utilization of a carefully curated image-text pair dataset. In the first training stage, LLaVA fine-tunes a multilayer perceptron (MLP) on the image-captioning task, aiming to map the output tokens from the image encoder into the language embedding space. In the second training stage, all network components, except for the pre-trained image encoder, are updated to optimize the model's instruction-following capabilities. Despite its strong performance in visual question answering (VQA), LLaVA lacks support for instructions in the form of speech. Many studies have also explored the direct integration of audio information processing into multimodal LLMs, such as ImageBind-LLM (Han et al., 2023) and Unified-IO 2 (Lu et al., 2024). However, there were fewer VLMs capable of supporting raw speech understanding until the recent introduction of GPT-4o (OpenAI, 2024), Gemini (Team et al., 2024) and VITA (Fu et al., 2024).

**Vision-Language-Action Model** A growing body of research has focused on applying VLMs in robotics, aiming to transfer general intelligence from software applications to the physical world. Specifically, two primary approaches have emerged for utilizing vision-language foundation models in the field of robot manipulation. One category of methods employs these foundation models only for high-level task planning, such as PaLM-E (Driess et al., 2023), SayCan (Ahn et al., 2022) and Code as Policies (Liang et al., 2023). In such studies, robots are typically equipped with pre-trained primitive skills, while the VLM is responsible for organizing these low-level skills to accomplish the target task. The other approach, exemplified by models such as RT-2 (Brohan et al., 2023), Roboflamingo (Li et al., 2024b), and OpenVLA (Kim et al., 2024), seeks to generate robot actions directly by fine-tuning the VLM with robot manipulation data. These models are commonly referred to as vision-language-action (VLA) models (Ding et al., 2024; Tong et al., 2024; Yue et al., 2024; Zhang et al., 2025). However, current VLA models typically focus on processing only two input modalities: textual instructions and visual observations (Belkhale et al., 2024). Some studies have also explored integrating additional input modalities, such as haptics and depth information, to further enhance model performance (Cai et al., 2024; Zhen et al., 2024). Although MUTEX (Shah et al., 2023) provides a unified policy for multimodal task specifications, it does not fully leverage the capabilities of recent vision-language models.

Nevertheless, few studies have investigated how speech modality inputs could be incorporated into VLA models. The most common approach to enabling speech input is to convert speech to text using an external speech recognition tool. However, this approach is not only complex but also results in the loss of auxiliary information present in the speech. To that end, an increasing body of research has recently started to explore the direct integration of speech into large language models in an end-to-end manner (Fu et al., 2024). Thus, our work takes a step further by developing a VLA model that supports speech instructions, showcasing how speech modality input enhances performance in scenarios where personalized knowledge is required.

## 3 METHOD

We present VLAS, a VLA model directly supporting speech instructions for robot manipulation. As illustrated in Figure 2, we first provide an overview of the VLAS architecture (Section 3.1). Section 3.2 introduces the curated SQA and CSI datasets, which are employed to train the VLAS model. Finally, in Section 3.3, we detail the training paradigm for speech instruction tuning.

### 3.1 ARCHITECTURE OF VLAS

**Overall Framework** VLAS takes human speech instructions $s$ and visual observations $\mathbf{O}$ as input to directly generate robot actions $\mathbf{a}$. The input image and speech instruction represented by frequency

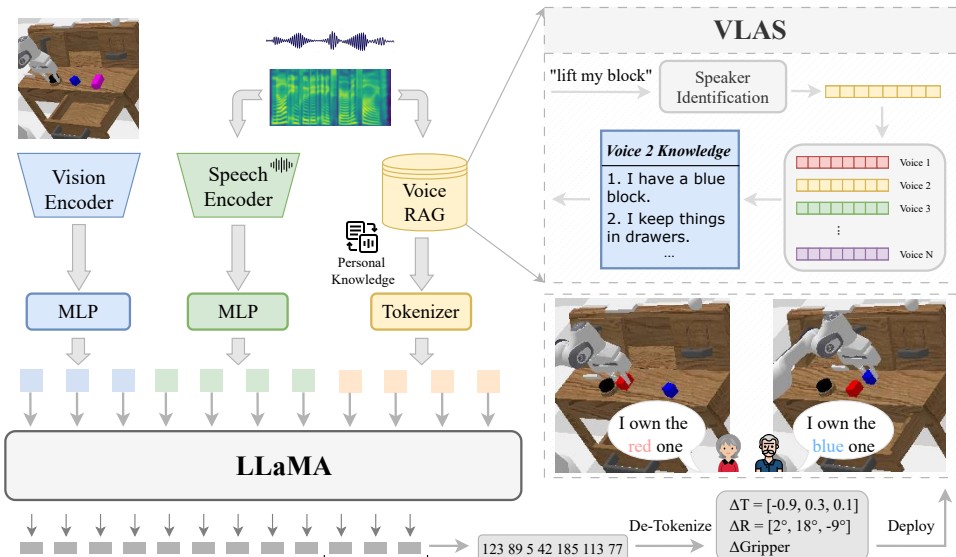

Figure 2: **Overall Framework of VLAS.** VLAS encodes visual and speech inputs via encoders and MLP layers to obtain respective embeddings. The Voice RAG module retrieves personalized knowledge based on speaker identification and converts it into embeddings using a text tokenizer. All embeddings are then processed by LLaMA to generate action tokens, which are subsequently detokenized into continuous values to control the robot's movements.

domain features are each converted into a sequence of embedding tokens through their correspond-ing encoders. During the inference phase, the output $\mathrm{RAG(s)}$ of the voice retrieval-augmented generation module is also tokenized into a sequence of embedding tokens. Both visual and speech tokens are transferred to separate MLPs to map them into the same language space. Subsequently, all the embedding tokens are concatenated as input to the LLM backbone. Formally:

$$\mathrm{Emb}(s, \mathbf{O}) = \mathrm{concat}(\mathrm{MLP}_s(\mathrm{Emb}_s(s)), \mathrm{Tok}_l(\mathrm{RAG}(s)), \mathrm{MLP}_v(\mathrm{Emb}_v(\mathbf{O}))), \qquad (1)$$

where $\mathrm{Emb}_s$, and $\mathrm{Emb}_v$ denotes speech and vision encoder, respectively; $\mathrm{MLP}_s$ and $\mathrm{MLP}_v$ means corresponding projector; $\mathrm{Tok}_l$ is the text tokenizer. This concatenated embedding is then fed into the LLM backbone to produce the predicted actions in an autoregressive manner as:

$$p(\mathbf{a} \mid \mathrm{Emb}(s, \mathbf{O})) = \prod_{i=1}^{N} p(a_i \mid \mathrm{Emb}(s, \mathbf{O}), a_{<i}) \qquad (2)$$

where $N$ denotes the number of dimensions for a single step action, and $\mathbf{a}$ is the discretized action tokens, which require a detokenizer to be converted into continuous values.

**Network Backbone** VLAS is built upon the vision-language model LLaVA, as illustrated in Fig-ure 2. In addition to the LLaMA LLM backbone, the key components of LLaVA are the Vision Transformer (ViT) (Dosovitskiy et al., 2021), which converts input image patches into a sequence of embedding tokens, and MLPs that map these tokens to the same semantic space as the LLM. When the vision tokens and text tokens are fed in together, the LLM can correlate these inputs and generate a corresponding response. In particular, we use the CLIP (Radford et al., 2021) model as the visual encoder and Vicuna (Chiang et al., 2023), a fine-tuned variant of LLaMA, as the foundation model.

**Speech Encoder** To equip our model with the ability to process speech modality input, we employ the Whisper (Radford et al., 2023) encoder $\mathrm{Emb}_s$ to convert a speech instruction $s$ into a sequence of hidden states $\mathrm{Emb}_s(s)$, similar to the visual tokens. Before being fed into the Whisper encoder, the speech signal is first transformed into an 80-bin mel-spectrogram using short-time Fourier transform (STFT) and then padded to a fixed length of 3000 frames. The speech encoder processes this mel-spectrogram and produces a sequence of 1500 hidden representations. Given that a long sequence of

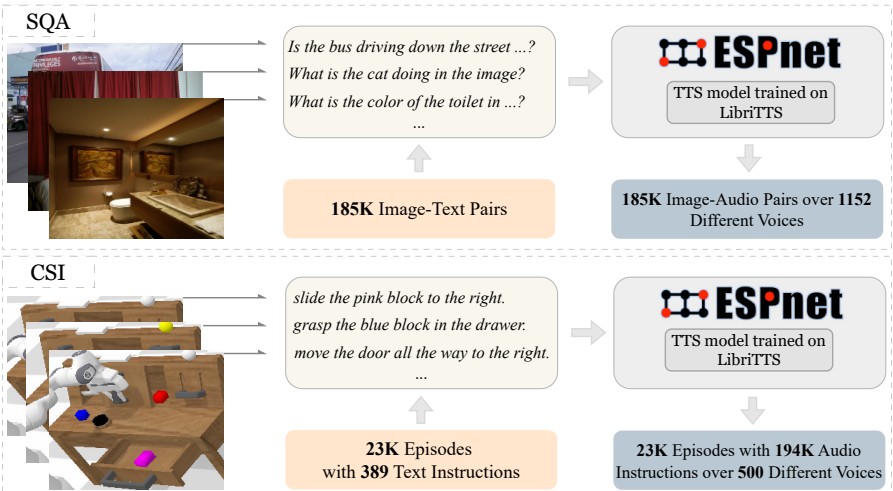

Figure 3: Data collection process for the SQA and CSI datasets.

speech tokens may impose a significant computational burden when directly input into the LLM, we apply a simple reshape operation along the time dimension, using a reduction factor of 5. An MLP is used to project the speech tokens into the semantic space shared with the text and vision tokens.

**Voice RAG** Retrieval-Augmented Generation (RAG) (Zhao et al., 2024) is a highly effective method for equipping large language models with the capability to efficiently process dynamic and up-to-date information. Human-spoken instructions frequently exhibit informality and lack of structure, resulting in inadequate semantic content for task completion. To address this issue, we propose a novel Voice RAG framework to bolster model performance on tasks that require extensive personal knowledge. The Voice RAG module allows our model to access additional customized knowledge beyond the original instruction content. As illustrated in Figure 2, the raw speech command is processed by the speaker identification module to extract a voiceprint. This voiceprint serves as a key to query an external database, retrieving relevant information. The retrieved data is then integrated as background knowledge and passed to the LLM, in conjunction with visual and speech tokens. To streamline this process, we utilize a pre-trained voiceprint extraction module, avoiding the need for from-scratch training. The integration of the Voice RAG significantly enhances the model's ability to comprehend and execute complex spoken commands by supplementing additional contextual information.

**Action Tokenization** We discretize a continuous action value into 256 uniformly spaced bins and represent them as integer indices. Specifically, we reutilize the 256 least frequently used tokens in the LLM vocabulary to serve as action tokens. Then, the robot action tokens across all motion dimensions can be concatenated with a space character to form a textual string, which serves as the training label. Consequently, a 7-dimensional action value is formatted as:

$$[x, \ y, \ z, \ \phi, \ \theta, \ \psi, \ g], \tag{3}$$

where $x, y, z$ represent the Cartesian coordinates of the end effector's position, $\phi, \theta, \psi$ denote the rotation angles of the end effector along each axis, and $g$ is the gripper state.

### 3.2 DATA COLLECTION FOR VLAS

Since traditional datasets used for fine-tuning VLM or VLA models do not include speech instructions, we constructed two new datasets to train our proposed model.

**Speech Question Answering (SQA) Dataset** The original visual instruction tuning dataset used for LLaVA contains extensive image-text question answering pairs, covering conversations, detailed descriptions, and complex reasoning tasks. Among the three aforementioned task types, the conversation subset follows a multi-turn format, whereas the others are single-turn. To construct the SQA dataset, we randomly sampled one round of dialogue from the multi-turn conversation subset and converted the textual questions into corresponding speech as shown in Figure3. These speech

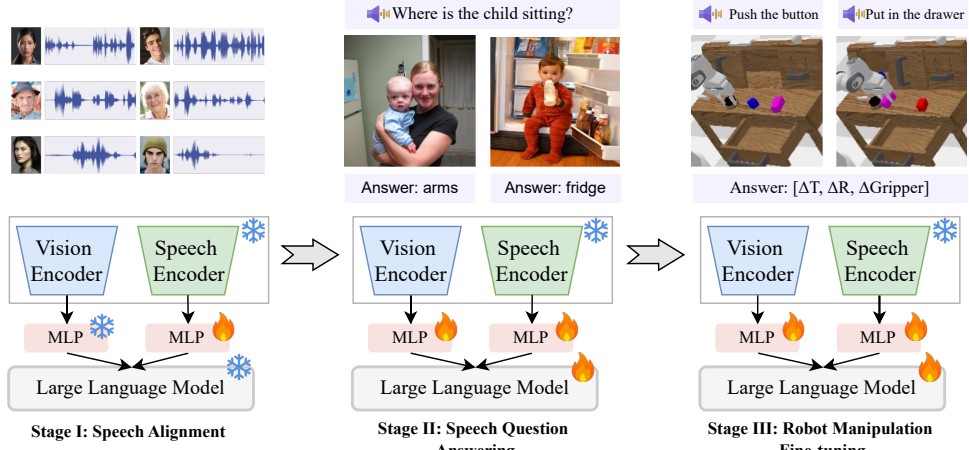

Figure 4: **Training paradigm of VLAS.** The training process of VLAS is divided into three stages. Stage I: Speech Alignment, where the model aligns speech with text through MLP fine-tuning. Stage II: Speech Question Answering, where the model is trained on both speech and visual question answering tasks to facilitate comprehension of multimodal inputs. Stage III: Robot Manipulation Fine-tuning, where the model is further fine-tuned to execute robot manipulation tasks using both speech and text instructions.

instructions, paired with associated images and textual answers, form the SQA dataset. We used the text-to-speech (TTS) tool ESPnet Hayashi et al. (2020) to generate the speech, specifically employing the pre-trained VITS TTS model Kim et al. (2021) trained on the LibriTTS dataset Zen et al. (2019), which supports over 2,000 distinct voices. During the conversion of textual questions to speech, the speaker's voice was randomly selected. In total, 185K SQA samples were generated, featuring over 1,152 different voices.

**CALVIN with Speech Instructions (CSI) Dataset** Given that conventional robot manipulation datasets contain only textual task instructions, we utilized the aforementioned TTS model to generate the corresponding speech instructions. For the CALVIN dataset, which contains 389 textual instructions, we employed 500 different voices to convert each instruction into speech, resulting in approximately 194K audio samples. In the training process, the raw robot manipulation datasets are structured as pairs of $((\text{Image}_t, \text{Instruction}_{text}), \text{Action}_t)$. To enable the robot policy model to support both text and speech instructions, we randomly replaced half of the training samples with the synthesized speech instructions.

## 3.3 TRAINING PARADIGM OF VLAS

The training process of VLAS consists of three stages, as depicted in Figure 4. The details of each stage are outlined below.

**Stage I: Speech Alignment** *The first stage focuses on coarse-grained modality alignment between speech and text*, achieved by fine-tuning the model using the LibriSpeech-360 speech recognition dataset Panayotov et al. (2015). During this phase, only the MLP layer between the speech encoder and the LLM backbone is updated to fulfill speech recognition tasks. It is worth mentioning that the speaker identification module for voiceprint extraction can be co-trained during this stage. However, this is optional, as we may directly employ a pre-trained speaker identification model.

**Stage II: Speech Question Answering Fine-tuning** *The second stage focuses on further enhancing the model's capability to process information from multiple input modalities.* At this stage, the model is fine-tuned using both our curated speech question answering (SQA) dataset and the original visual question answering (VQA) datasets from LLaVA, as well as the LibriSpeech-100 speech recognition dataset Panayotov et al. (2015). Throughout this phase, all network components are updated, with the exception of the pre-trained image and speech encoders. Notably, after this stage, we obtain the foundation model, referred to as VLAS-Base, for the subsequent robot manipulation task. The

VLAS-Base model can also serve as a valuable resource for the research community in advancing studies on multimodal large language models.

**Stage III: Robot Manipulation Fine-tuning** *In the final training stage, the model is fine-tuned on the CSI robot manipulation dataset in a manner similar to that of stage 2*. Each sample in this dataset contains a complete motion trajectory, represented as a sequence of robot actions, along with visual observations from two distinct views and corresponding human instructions in either textual or speech form. For simplicity, the two images at each time step are concatenated together.

# 4 EXPERIMENTS AND RESULTS

In this section, we conduct a series of experiments to assess the effectiveness of the proposed method from multiple perspectives. Section 4.1 first provides a quantitative evaluation of the performance of our VLAS model on the CALVIN benchmark. In Section 4.2, we then constructed a new benchmark consisting of various customized tasks to further assess our method. Finally, in Section 4.4, to verify whether our foundation model for robot manipulation truly understands speech instructions without compromising LLaVA's original performance, we evaluate VLAS-Base on general multimodal benchmarks, as well as two benchmarks for speech understanding.

## 4.1 ROBOT MANIPULATION WITH SPEECH INSTRUCTIONS

To quantitatively assess the performance of our proposed model for robot manipulation tasks, we conduct experiments on the CALVIN benchmark, which comprises 1,000 long-horizon tasks. Each long-horizon task consists of a sequence of five successive sub-tasks, accompanied by a corresponding human command. We trained a traditional VLA model with the same configurations by directly fine-tuning the LLaVA backbone, without support for speech instructions, as the baseline.

Table 1: Performance of different robot policy models on the CALVIN benchmark. [+]: Evaluated with the ground truth textual instructions. [*]: Evaluated with the speech instructions. On this benchmark, the Voice RAG module is not utilized by VLAS to acquire any customized knowledge.

| Models | Splits | LH-1 | LH-2 | LH-3 | LH-4 | LH-5 | Len |
|---|---|---|---|---|---|---|---|
| MCIL[+] | ABCD/D | 37.3% | 2.7% | 0.2% | 0.0% | 0.0% | 0.40 |
| HULC[+] | ABCD/D | 89.2% | 70.1% | 54.8% | 42.0% | 33.5% | 2.90 |
| RT-1[+] | ABCD/D | 84.4% | 61.7% | 43.8% | 32.3% | 22.7% | 2.45 |
| VLA[+] | ABCD/D | *95.5%* | *85.0%* | *74.9%* | *66.8%* | *58.2%* | *3.80* |
| VLAS[+] | ABCD/D | *94.5%* | *84.4%* | *73.6%* | *64.6%* | *56.6%* | *3.74* |
| Roboflamingo[*]+ASR | ABCD/D | 89.8% | 78.6% | 68.2% | 56.5% | 48.3% | 3.41 |
| VLA[*]+ASR | ABCD/D | 88.7% | 74.1% | 61.0% | 49.2% | 40.2% | 3.13 |
| VLAS[*] | ABCD/D | **94.2%** | **84.0%** | **73.2%** | **64.3%** | **54.6%** | **3.70** |
| VLAS[*](Real) | ABCD/D | 93.6% | 82.8% | 71.6% | 61.4% | 51.3% | 3.61 |

As shown in Table 1, our VLAS, with either textual or speech instructions, significantly outperforms the official MCIL Lynch* & Sermanet* (2021) model and other prevalent models such as HULC Mees et al. (2022) and RT-1 Brohan et al. (2022). Specifically, VLAS with textual instructions also achieves performance comparable to the baseline VLA model. Moreover, our VLAS is compared for speech modality input with the baseline VLA model and another powerful VLA model, Roboflamingo, both similarly derived from the VLM. Since traditional robot policy models do not directly support speech instructions, we employ an external ASR model to transcribe the speech instructions into text. The most powerful ASR model, Whisper large-v2, released by OpenAI is used in the experiments. To generate the speech instructions for evaluation, the previously discussed TTS model is employed with 39 novel voices not included in the SQA and CSI datasets. For each instruction, a corresponding voice is randomly sampled. In addition, we have included real speech instructions recorded from 10 individuals for evaluation. As can be observed, even with real speech instructions, our VLAS still achieves strong performance, only slightly behind the VLA baseline with a gap of 0.19.

We found that VLAS significantly outperforms the other two methods that utilize a cascading pipeline for speech understanding. We attribute this to the higher accuracy of our method in recognizing speech instructions for robot manipulation, as the model has been fine-tuned on a specialized dataset. Conversely, the external ASR model is less sensitive to controlling commands for the robot, leading to amplified propagation errors. It is important to highlight our method is orthogonal to other VLA models, and thus, can be combined with them to achieve superior performance.

## 4.2 Robot Manipulation for Customized Tasks

Table 2: Performance of three types of customized tasks for robot manipulation. [+]: Evaluated with the ground truth textual instructions. [*]: Evaluated with the speech instructions. On this benchmark, the Voice RAG module is utilized by VLAS to acquire customized knowledge.

| Models | Ownership | Preference | Compound | Compound-Multistage | | Avg. |
|---|---|---|---|---|---|---|
| | | | | Stage-1 | Stage-2 | |
| VLA$^+$ | 17.9% | 30.8% | 23.1% | 35.9% | 5.1% | 19.2% |
| VLAS$^*$ | 94.7% | **84.6%** | **100.0%** | **100.0%** | **66.7%** | **86.5%** |
| VLAS$^*$(Real) | 89.5% | 70.0% | 100.0% | 90.0% | 55.0% | 78.6% |
| VLAS$^*$−RAG | 15.4% | 12.8% | 25.6% | 33.3% | 10.3% | 16.0% |
| VLA$^+$+RAG | **97.4%** | 84.6% | 97.4% | 82.1% | 48.7% | 82.0% |

To evaluate our model's capability in executing personalized tasks, we developed a new benchmark comprising diverse, unstructured spoken instructions within the simulation environment. All these tasks require the robot to utilize personal knowledge beyond superficial semantic content. Particularly, this benchmark includes three task categories: (1) Object Ownership Tasks: The robot must interact with the appropriate objects according to their ownership. When given a spoken instruction, the robot needs to identify the person's intention and use the correct object belonging to them. (2) User Preference Tasks: These tasks necessitate the robot to comprehend the user's preferences. Given the identical command, the robot is expected to perform different actions depending on the specific user's preferences. (3) Compound Tasks: Tasks in this category require the robot not only to select appropriate objects, but also to perform actions that align with the user's preferences. In particular, this category includes multistage tasks where the robot is required to respond to two successive human instructions. Since the outcome of the previous task can easily impact the execution of the subsequent task, these multistage tasks pose a greater challenge. For each task category, there are a total of 39 unseen voices beyond training datasets.

Table 2 presents a detailed comparison between the VLA baseline and VLAS. Because the VLA baseline relies solely on text instructions and lacks access to background knowledge, its performance is severely limited, with an average success rate of below $20\%$. Such a model can only perform tasks through random attempts or by drawing inferences from contextual information. However, VLAS, which directly receives raw speech input, leverages the Voice RAG to access individual-specific knowledge, allowing it to perform customized operations more effectively. As a result, our model demonstrates much better performance on this benchmark, achieving an average success rate of over $86\%$. Figure 5 and Figure 6 present several concrete case studies showing how the proposed method performs customized operations for different users.

We introduce real speech instructions for evaluation, which also demonstrates acceptable performance. Ablation studies are conducted to further validate the effectiveness of our proposed Voice RAG module. It can be seen from Table 2 that when the RAG module is removed, the performance of VLAS significantly degrades on the customized benchmark. Meanwhile, when our RAG module is integrated with the VLA, its performance significantly improves. Both of the ablation studies above demonstrate the effectiveness of the Voice RAG module.

## 4.3 Experiments with a Real-World UR5 Robot Arm

We fine-tune our VLAS-Base by utilizing both the Berkeley UR5 demonstration dataset and our own cup-picking dataset. This results in a VLAS model that can be deployed on real-world robots. As shown in Figure 7, our model can respond to different actions according to the personal information of the speaker, like picking up the specific cup considering the ownership.

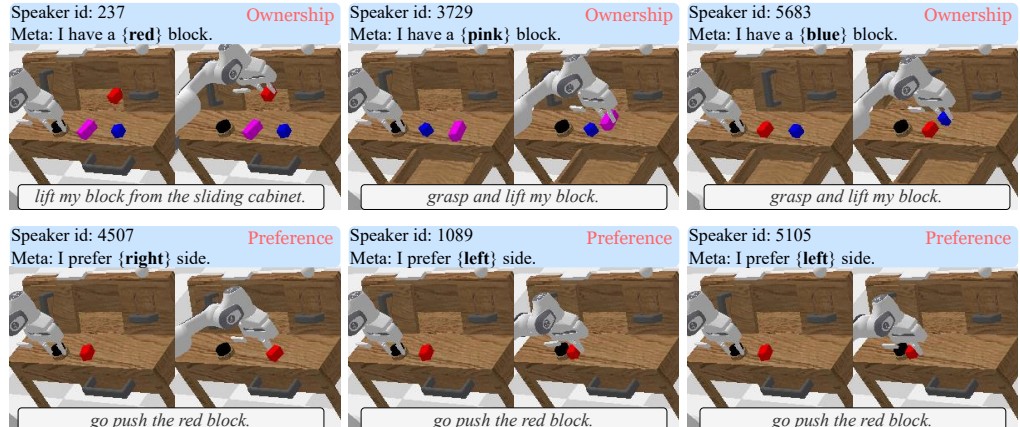

Figure 5: Demonstration of object ownership tasks (top row) and user preference tasks (bottom row) for customized robot manipulation.

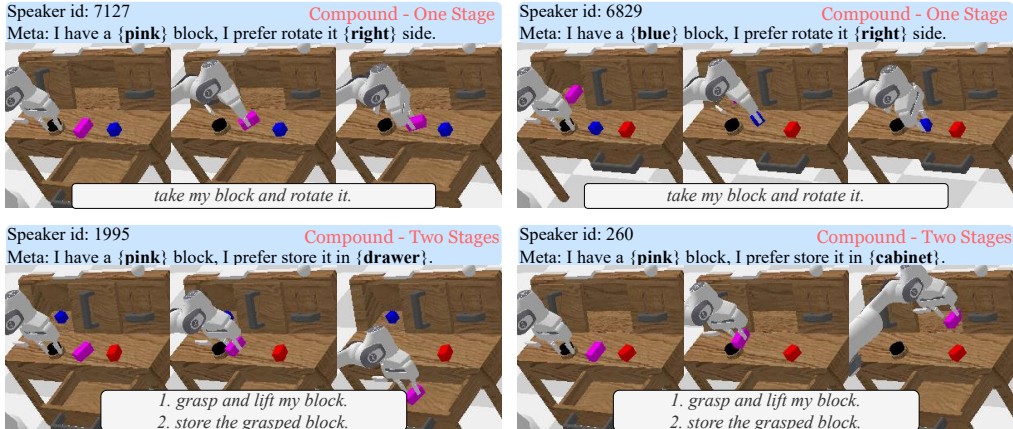

Figure 6: Demonstration of compound tasks for customized robot manipulation.

## 4.4 Analysis for the VLAS-Base Foundation Model

The multimodal understanding capability of the VLAS-Base is critical when fine-tuning it with robot trajectories to develop the proposed VLAS. Therefore, we quantitatively assess the performance of our VLAS-Base from two perspectives. First, the VLAS-Base is expected to achieve performance comparable to the original LLaVA model, as the ability to comprehend visual and language information serves as the foundation for intelligent robot manipulation.

Table 3 provides a detailed comparison between the VLAS-Base and other prevalent VLMs across general multimodal benchmarks. As can be observed, VLAS-Base obtains nearly the same performance to LLaVA, while significantly outperforming other VLMs. These results indicate that the introduction of the speech modality does not degrade the performance of the foundation model.

Second, the VLAS-Base model is also expected to have a strong understanding of speech modality input. For this purpose, we conduct experiments on the LibriSpeech automatic speech recognition benchmark and our self-constructed speech question answering benchmark, SGQA. Given the lack of Q&A evaluation benchmarks for image-speech pairs, we converted all textual questions in the GQA benchmark for visual question answering into speech format with an external TTS model, resulting in the SGQA benchmark. For the speech recognition benchmark, we employ the state-of-the-art Whisper large-v2 model as the baseline. For the speech question answering benchmark, since prevalent VLMs typically do not support speech input, we use LLaVA and BLIP-2 with ground-truth textual instructions as baselines.

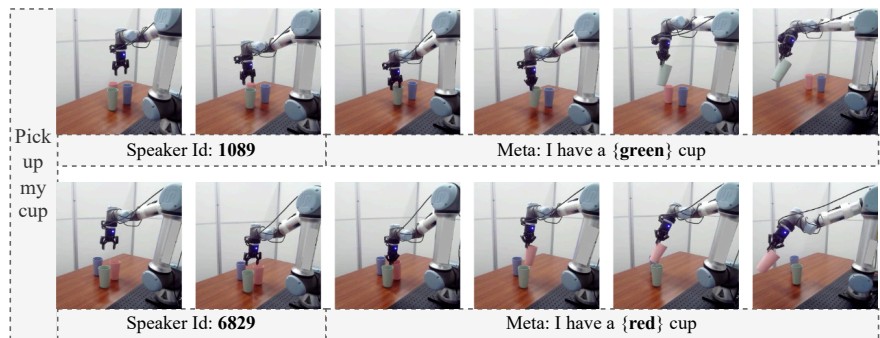

Figure 7: Demonstration of success cases of VLAS on the real-world UR5 robot arm.

In Table 4, VLAS-Base achieves comparable performance to Whisper large-v2 on the LibriSpeech test set. Considering that a reduction factor is applied to downsample the speech spectrum for VLAS-Base, its performance could potentially be improved by optimizing this factor or by employing a more advanced downsampling module. Moreover, although VLAS-Base falls behind LLaVA with ground-truth textual instructions on the SGQA benchmark, it still surpasses BLIP-2.

These results indicate that our foundation model, used for developing VLAS, can effectively process diverse speech instructions. We can even utilize co-training with robot trajectories and speech question answering data to further improve VLAS's capacity to handle more complex human commands.

Table 3: Performance comparison between state-of-the-art VLMs to VLAS-Base across diverse multimodal evaluation benchmarks.

| Model | LLM | VQA$^{v2}$ | VizWiz | SQA$^I$ | VQA$^T$ | POPE | GQA |
|---|---|---|---|---|---|---|---|
| BLIP-2 | Vicuna-13B | 65.0 | 19.6 | 61.0 | 42.5 | 85.3 | 41.0 |
| InstructBLIP | Vicuna-13B | - | 33.4 | 63.1 | 50.7 | 78.9 | 49.5 |
| Qwen-VL | Qwen-7B | **78.8** | 35.2 | 67.1 | 63.8 | - | 59.3 |
| LLaVA v1.5 | Vicuna-7B | **78.8** | 50.0 | 66.8 | **58.2** | **85.9** | **62.0** |
| VLAS-Base | Vicuna-7B | 78.7 | **51.1** | **72.2** | 58.1 | 85.5 | **62.0** |

Table 4: Performance comparison on LibriSpeech and SGQA benchmark, using word error rate (WER) and accuracy as evaluation metrics. LLaVA and BLIP-2 employ the ground truth textual insturctions on SGQA.

| Model | LibriSpeech (WER) | SGQA |
|---|---|---|
| LLaVA v1.5 | N/A | 62.0 |
| BLIP-2 | N/A | 41.0 |
| Whisper | 2.7% | N/A |
| VLAS-Base | 2.79% | 50.8 |

## 5 CONCLUSION

This paper presents an end-to-end VLA model for robot manipulation that is capable of understanding speech instructions without relying on an external speech recognition system. As the raw speech is directly taken as the model's input, auxiliary information in the speech, such as voiceprint, can be fully utilized to more effectively complete the given task. In particular, we introduce a Voice RAG method for our model to improve its performance in following spoken instructions that require extensive individual-specific knowledge. Consequently, the integration of speech modality data in VLAS not only simplifies the overall pipeline for robot control but also enables the robot to handle a wide range of customized tasks. Our future work may focus on exploring other auxiliary information in human speech or environmental sounds to enable the robot to better understand and complete complex tasks.

ACKNOWLEDGMENTS

This work was supported by the National Science and Technology Innovation 2030 - Major Project (Grant No. 2022ZD0208800), and NSFC General Program (Grant No. 62176215). In addition, we thank Gongsheng Li for his valuable discussions and assistance with data processing.

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

## A    TRAINING DETAILS

We perform fine-tuning in Stage I on the train-clean-100 split of the LibriSpeech dataset for 5 epochs, using a learning rate of 1e-3 and a batch size of 16. Subsequently, the fine-tuning in Stage II is conducted on our SQA dataset, along with the released LLaVA 665K instruction-following dataset and the train-clean-360 split of LibriSpeech, for 1 epoch using a learning rate of 2e-5 and a batch size of 16. Finally, we fine-tune the model on the CSI robot manipulation dataset for 1 epoch, with a learning rate of 2e-5 and a batch size of 16. Specifically, we combined actions from 5 time steps into a single training label to increase the operating frequency of the robot policy model. The Adam optimizer without weight decay and a cosine learning rate schedule with a 3% warmup ratio are used throughout the experiments. Flash Attention 2, BF16, and TF32 are enabled to achieve a balance between training speed and precision.

All models are trained using 8× A100 GPUs, except for the fine-tuning in Stage I. We empirically found that employing a single GPU for coarse-grained speech alignment yields better performance.

## B    EXTENDED EXPERIMENTAL RESULTS

### B.1    FAILURE CASES OF VLAS AND VLA ON THE CUSTOMIZATION BENCHMARK

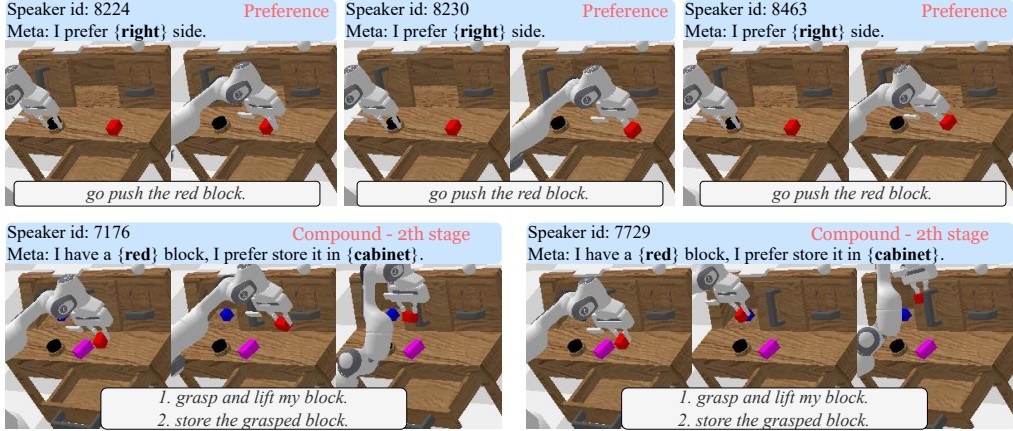

Figure 8: Demonstration of failure cases of VLAS on the customization benchmark.

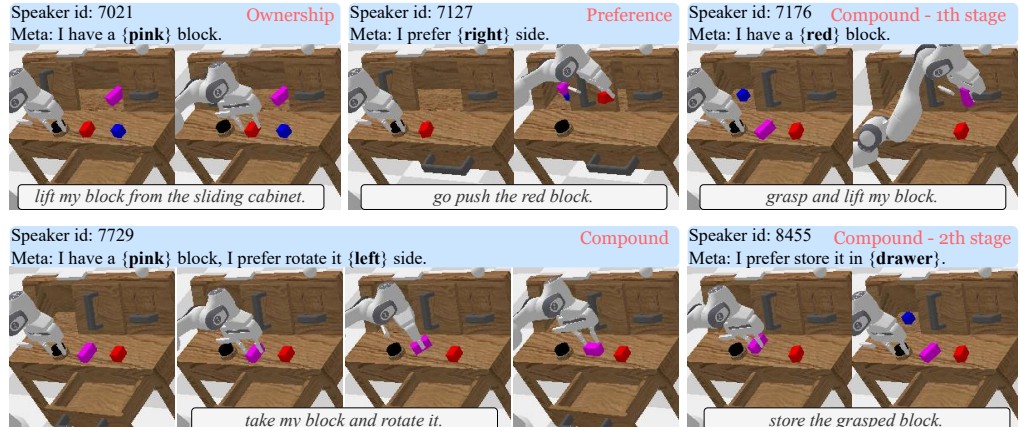

Figure 9: Demonstration of failure cases of VLA on the customization benchmark.

We conducted additional analysis on the failure cases of VLAS and VLA on the customization benchmark to better identify the underlying reasons. As observed in the Figure 8, failure cases of the VLAS model mainly occur in the preference task and the second phase of the compound task. The error pattern is more consistent, suggesting that the model understands the instructions but fails to execute the actions successfully. We conjecture this issue can be addressed by refining the policy model's architecture and training process. On the contrary, the VLA model exhibits a diverse range of error patterns, as illustrated in Figure 9. Since the VLA model has access only to superficial semantic information from human instructions, it relies on random attempts to complete these personalized tasks, leading to numerous failures.

## B.2    COMPARISON WITH ROBOFLAMINGO ON THE CALVIN BENCHMARK

RoboFlamingo is another prominent VLA model reported on the CALVIN Benchmark. Table 5 provides a comparison between VLAS and RoboFlamingo on the CALVIN Benchmark using textual instructions. It can be seen that VLAS performs slightly behind RoboFlamingo mainly due to lack of historical information when predicting actions. When the historical information, i.e. the LSTM policy head, is removed, the performance of RoboFlamingo significantly deteriorates. Thus, we can leverage similar approaches to further enhance the performance of our model, as these two methods are completely orthogonal.

Table 5: Comparison with RoboFlamingo on the CALVIN Benchmark. The performance of RoboFlamingo without historical information is derived from results presented in their original paper. [+]: Evaluated with the ground truth textual instructions.

| Models | Splits | LH-1 | LH-2 | LH-3 | LH-4 | LH-5 | Len |
|---|---|---|---|---|---|---|---|
| Roboflamingo[+] | ABCD/D | 96.4% | 89.6% | 82.4% | 74.0% | 66.0% | 4.09 |
| Roboflamingo[+](w/o Hist) | ABCD/D | ~60% | ~20% | ~20% | ~20% | ~20% | ~1.0 |
| VLAS[+] | ABCD/D | 94.5% | 84.4% | 73.6% | 64.6% | 56.6% | 3.74 |

## B.3    COMPARISON WITH OPENVLA ON THE CALVIN BENCHMARK

We further compare the performance of our model with both a pre-trained OpenVLA model and its fine-tuned variant on the CALVIN benchmark. For the pre-trained OpenVLA model, we directly utilized the "openvla-7b-finetuned-libero-10" checkpoint, as it has been tailored for the Franka Emika Panda in a simulation environment, closely resembling our experimental setup. For the fine-tuned OpenVLA model, we followed their officially recommended fine-tuning setup. The CALVIN dataset was converted into the unified RLDS format to enable the fine-tuning of OpenVLA, as they suggested.

It can be observed from Table 6 that the pre-trained OpenVLA model fails to achieve a zero shot generalization on the CALVIN benchmark. In addition, though the fine-tuned OpenVLA model

can complete some tasks in the initial stage of the long-horizon sequence, its overall performance remains suboptimal. We hypothesize that the poor performance of OpenVLA is due to its limitation of supporting only third-person view input. Visual feedback from the robot's end effector is of vital importance for facilitating precise manipulation tasks, such as object grasping and rotation. However, this functionality is absent in the current OpenVLA.

Table 6: Comparison with OpenVLA on the CALVIN Benchmark. [+]: Evaluated with the ground truth textual instructions.

| Models | Splits | LH-1 | LH-2 | LH-3 | LH-4 | LH-5 | Len |
|---|---|---|---|---|---|---|---|
| Pre-trained OpenVLA[+] | ABCD/D | 0.0% | 0.0% | 0.0% | 0.0% | 0.0% | 0 |
| Fine-tuned OpenVLA[+] | ABCD/D | 30.4% | 2.6% | 0.4% | 0.1% | 0.0% | 0.34 |
| VLAS[+] | ABCD/D | 94.5% | 84.4% | 73.6% | 64.6% | 56.6% | 3.74 |

### B.4 EXPERIMENTAL EVALUATION ON THE CALVIN BENCHMARK USING ABC/D SPLITS

To better evaluate our model's generalization capability to novel scenes, we conducted experiments in which the model was trained on ABC splits and tested on the D split. It can be observed that, despite all models experiencing performance degradation due to the domain gap, our VLAS achieved performance comparable to RoboFlamingo while outperforming the other models.

Table 7: Performance of different robot policy models on the CALVIN benchmark. [+]: Evaluated with the ground truth textual instructions. [*]: Evaluated with the speech instructions. On this benchmark, the Voice RAG module is not utilized by VLAS to acquire any customized knowledge.

| Models | Splits | LH-1 | LH-2 | LH-3 | LH-4 | LH-5 | Len |
|---|---|---|---|---|---|---|---|
| MCIL[+] | ABC/D | 30.4% | 1.3% | 0.2% | 0.0% | 0.0% | 0.31 |
| HULC[+] | ABC/D | 41.8% | 16.5% | 5.7% | 1.9% | 1.1% | 0.67 |
| RT-1[+] | ABC/D | 53.3% | 22.2% | 9.4% | 3.8% | 1.3% | 0.9 |
| VLA[+] | ABC/D | 83.1% | 58.4% | 34.7% | 23.1% | 15.1% | 2.14 |
| Roboflamingo[+] | ABC/D | 82.4% | 61.9% | 46.6% | 33.1% | 23.5% | **2.48** |
| VLAS[+] | ABC/D | 85.9% | 59.2% | 38.5% | 25.9% | 17.6% | 2.27 |
| VLA[*]+ASR | ABC/D | 74.7% | 54.1% | 38.4% | 24.1% | 16.5% | 2.04 |
| VLAS[*] | ABC/D | **87.2%** | **64.2%** | **40.9%** | **28.1%** | **19.6%** | **2.40** |

Moreover, we conducted similar experiments on our personalization benchmark. The results demonstrate that our model is capable of handling novel scenes.

Table 8: Performance of three types of customized tasks for robot manipulation. [+]: Evaluated with the ground truth textual instructions. [*]: Evaluated with the speech instructions. On this benchmark, the Voice RAG module is utilized by VLAS to acquire customized knowledge.

| Models | Ownership | Preference | Compound | Compound-Multistage | | Avg. |
|---|---|---|---|---|---|---|
| | | | | Stage-1 | Stage-2 | |
| VLA[+] | 20.5% | 5.1% | 0.0% | 10.3% | 0.0% | 6.4% |
| VLAS[*] | 64.1% | 61.5% | 87.2% | 74.4% | 7.7% | 55.1% |
| VLAS[*]−RAG | 15.4% | 23.1% | 0.0% | 12.8% | 0.0% | 9.6% |
| VLA[+]+RAG | 82.1% | 71.8% | 84.6% | 82.1% | 10.3% | 62.2% |

### B.5 INFERENCE EFFICIENCY ANALYSIS

This paper employs two key optimizations to enhance the inference speed of VLAS: downsampling the speech spectrogram and implementing an action update strategy with multi-step prediction and execution. Speech spectrogram downsampling is a widely used strategy to accelerate speech signal processing, where adjacent x-frame spectrograms are aggregated into a single-frame feature through a reshaping operation, effectively reducing the time dimension length. In our experiments, we used

the x = 5. Since the effectiveness of this approach has been validated in numerous speech recognition and generation tasks, we did not perform additional related analyses. Given that the state of the environment typically does not change significantly over a short period, our work adopts a simple yet effective multi-step prediction and execution policy. Specifically, we set the number of steps for both VLA and VLAS to r=5. As shown in Table 9, when r=5 , both the VLA and VLAS models achieve significant speedups while also demonstrating improved performance on the CALVIN benchmark.

Table 9: Inference efficiency of different models and their average performance on the CALVIN benchmark. [+]: Evaluated with the ground truth textual instructions. [*]: Evaluated with the speech instructions.

| Models | Actions / Sec (Hz) | Len |
|---|---|---|
| VLA[+](r=1) | 1.89 | 2.30 |
| VLAS[*](r=1) | 1.17 | 2.02 |
| VLA[+](r=5) | 3.60 | 3.80 |
| VLAS[*](r=5) | 2.50 | 3.70 |

Table 10: Inference efficiency of different models and their average performance on the CALVIN benchmark. [+]: Evaluated with the ground truth textual instructions. [*]: Evaluated with the speech instructions.

| Models | Actions / Sec (Hz) | Len |
|---|---|---|
| VLAS[*](r=1) | 1.17 | 2.02 |
| VLAS[*](r=5) | 2.50 | 3.70 |
| VLAS[*](r=12) | 2.88 | 3.35 |
| VLAS[*](r=20) | 3.80 | 0.70 |

We supplemented our results with an analysis of the inference speed and performance of VLAS across different values of r. Table 10 indicates that r=5 achieves an optimal balance between inference efficiency and manipulation performance.

