# OpenReview forum: "VLAS: Vision-Language-Action Model with Speech Instructions for Customized Robot Manipulation"
_ICLR.cc/2025/Conference — ICLR 2025 Poster_

### Official Review · Reviewer_5GYV · 2024-10-31

**Soundness:** 3
**Presentation:** 3
**Contribution:** 3
**Rating:** 6
**Confidence:** 3

**Summary:**

This work proposes VLAS and integrates speech for robot manipulation. It makes communication more natural and presents two new datasets for the community.

**Strengths:**

1. This paper is easy to read and follow.
2. The core idea and motivation are interesting and practical. I believe it is great to have the speech modality in interactions with robots, as it feels natural and intuitive. Using speech to capture human intent is also a good idea. Creating a unified model provides a solid foundation for future work.
3. The illustrations are very helpful.
4. The datasets provided to the community are beneficial for the field.

**Weaknesses:**

Some ablation studies may be needed. Please refer to the "Questions" section.

**Questions:**

1. Are there any statistics from real-world experiment results? I think it's important to understand how the impressive results from simulations can transfer to real-world applications.
2. You use the RAG and the pretrained speech model to obtain the voiceprint, which is then used to retrieve preferences from the database. It seems the voiceprint is derived from the RAG module. You only present the VLA experiment in a customized task, and I wonder about the performance of the VLA combined with speech RAG in this context. I think this need to be given to support your motivation ( why not use the cascading pipeline, you mentioned the cost and the intent capture), I suspect that the speech RAG can capture the intend and give the intend text from the database to VLA model.
3. I think you need some more ablation studies, such as examining the effects of not using the RAG module in the customized task.

I think if these questions are addressed, I will raise my score cause I think the unified large model for speech, vision, text and action is meaningful.

---

> ### Author Response · Authors · 2024-11-24
> **Thank you very much for your affirmative review of our work. We will explain your concerns point by point.**
>
> **Q1**: `Are there any statistics from real-world experiment results? I think it's important to understand how the impressive results from simulations can transfer to real-world applications.`
>
> **A1**: We agree that incorporating statistics from real-world experiments would enrich the content of the paper. We are currently working on this and will include the relevant results as soon as they are available.
>
> **Q2**: `You use the RAG and the pretrained speech model to obtain the voiceprint, which is then used to retrieve preferences from the database. It seems the voiceprint is derived from the RAG module. You only present the VLA experiment in a customized task, and I wonder about the performance of the VLA combined with speech RAG in this context. I think this need to be given to support your motivation.`
>
> **A2**: As you summarized, our motivation is to enable the processing of raw speech instructions through a unified end-to-end model, aiming to simplify model architecture and reduce the impact of potential cumulative errors. Additionally, the introduction of raw speech enables a more intuitive extraction of voiceprints, allowing the voice RAG to effectively capture individual intents.
>
> To better analyze the role of the RAG module in our method, we can indeed integrate it with the traditional VLA model, as you suggested. The relevant results are presented in the following table. In this setup, the RAG module can also convey individual intentions to the VLA model, thereby improving its performance on the personalization benchmark.
>
> | Model | Ownership | Preference | Compound | Compound-Multistage (Stage-1) | Compound-Multistage(Stage-2) | Avg. |
> | --- | --- | --- | --- | --- | --- | --- |
> | VLA              | 17.9% | 30.8% | 23.1%   | 35.9%   | 5.1%   | 19.2% |
> | VLAS            | 94.7% | 84.6% | 100.0% | 100.0% | 66.7% | 86.5% |
> | VLA   + RAG | 97.4% | 84.6% | 97.4% | 82.1% | 48.7% | 82.0% |
> | VLAS - RAG | 15.4% | 12.8% | 25.6% | 33.3% | 10.3% | 16.0% |
>
> **Q3**: `I think you need some more ablation studies, such as examining the effects of not using the RAG module in the customized task.`
>
> **A3**: Thank you very much for your valuable suggestions. We have included the corresponding experiments, as shown in the table above.

---

> > ### Comment · Reviewer_5GYV · 2024-11-25
> >
> > I found that in the experiment (training on dataset ABC and testing on D), the results show that VLA+RAG outperforms VLAS by about 7%. Does this indicate that the generalization ability of VLAS is weaker compared to the cascading pipeline of VLA+RAG? If the cascading pipeline demonstrates better performance in identity-related tasks, it seems there might not be a strong need to use models like VLAS. What are your thoughts on this?

---

> > > ### Author Response · Authors · 2024-11-26
> > > **Thank you very much for your quick reply!**
> > >
> > > **Q1**:`I found that in the experiment (training on dataset ABC and testing on D), the results show that VLA+RAG outperforms VLAS by about 7%. Does this indicate that the generalization ability of VLAS is weaker compared to the cascading pipeline of VLA+RAG? If the cascading pipeline demonstrates better performance in identity-related tasks, it seems there might not be a strong need to use models like VLAS. What are your thoughts on this?`
> > >
> > > **A1**:
> > >
> > > **a**. Our generalization experiments on the ABC→D task consist of two parts: one conducted on the standard CALVIN benchmark and the other on our customized benchmark. Despite the VLA + RAG achieves a better performance on the customized benchmark, the VLA itself perfoms not as well as VLAS on the CALVIN benchmark as in the table. Thus, we may not conclude that VLAS is definitively weaker in terms of generalization, especially considering that the CALVIN benchmark comprises a total of 1,000 long-horizon tasks.
> > > | Model (ABC→D) | LH-1 | LH-2 | LH-3 | LH-4 | LH-5 | Avg. Len |
> > > | --- | --- | --- | --- | --- | --- | --- |
> > > | VLA | 0.747 | 0.541 | 0.384 | 0.241 | 0.165 | 2.04 |
> > > | VLAS (Speech) | 0.872 | 0.642 | 0.409 | 0.281 | 0.196 | 2.40 |
> > >
> > > **b**. Moreover, it is worth noting that in the customization experiments, the VLA is employed with ground truth textual instructions for simplicity. However, in realistic deployment, the VLA needs to be paired with an additional ASR model, which may lead to significant reductions in inference speed and potential performance degradation, as shown below.
> > > | Model | Actions / Sec (Hz) | Avg. Len |
> > > | --- | --- | --- |
> > > | VLA (r=5) | 3.60 | 3.80 |
> > > | ASR + VLA (r=5) | 0.17 | 3.13 |
> > > | VLAS (r=5) | 2.50 | 3.70 |
> > >
> > > **c**. For both of the above reasons, we believe that developing an end-to-end robotic policy model incorporating speech processing remains valuable. Even though in some cases, the relevant performance may not be satisfactory, however, there will always be relevant techniques for its further improvement, such as interleaved fine-tuning in VILA [1] and end-to-end speech language models like GPT-4o [2] and VITA [3].
> > >
> > > **Ref.**
> > >
> > > [1] Lin Ji, et al. “VILA: On Pre-training for Visual Language Models.” CVPR 2024.
> > >
> > > [2] https://openai.com/index/hello-gpt-4o/
> > >
> > > [3] Fu, Chaoyou, et al. "VITA: Towards Open-Source Interactive Omni Multimodal LLM." arXiv preprint arXiv:2408.05211 (2024).
> > >
> > > **Q2**: `Are there any statistics from real-world experiment results?`
> > >
> > > **A2**: We have expanded our experiments to include two types of tasks using a real UR5 robotic arm to provide additional statistics about VLAS.
> > >
> > > **1. Ownership**
> > > | Blue Cup | Green Cup | Red Cup | Avg. |
> > > | --- | --- | --- | --- |
> > > | 9/10 | 8/10 | 9/10 | 86.67% |
> > >
> > > **2. Compound**
> > > | Blue Bowl to Grey Pot | Blue Bowl to Red Pot | Green Bowl to Grey Pot | Green Bowl to Red Pot | Avg. |
> > > | --- | --- | --- | --- | --- |
> > > | 7/10 | 7/10 | 4/10 | 5/10 | 57.5% |
> > >
> > > The first type of task is the ownership task, where the robot is instructed to select the correct cup for three different individuals based on the command, “Please pick up my cup.” The second type of task is a compound task that combines both ownership and preference considerations. Four different instructors give the command “Pick my smaller bowl into my pot.” To ensure efficiency while maintaining consistency in experimental conditions, we set the total number of rollouts per subtask to 10, as preparing a single rollout (including scene transitions and other adjustments) takes approximately 5 minutes on average.
> > >
> > > We found that VLAS can successfully transfer to real-world scenarios for most manipulation tasks. We did not compare it with a VLA baseline due to the reason that when VLA works independently without any background knowledge, it fails to produce meaningful actions for our customized tasks. The corresponding results, can be speculated from the simulation experiments, appear to exhibit characteristics similar to random behavior.

---

> > > > ### Comment · Reviewer_5GYV · 2024-11-26
> > > >
> > > > Thank you for your response and the experiment. However, I still have a question about the metric: how do you calculate actions per second? I think that ASR should be only used at first and doesn't need to be repeated again and again. Therefore, in real deployments, maybe the main concern for the inference speed is the delta time of two actions, is the action per second you provided computed like this?

---

> > > > > ### Author Response · Authors · 2024-11-28
> > > > > **Thank you for your insightful question.**
> > > > >
> > > > > Thank you for your insightful question. Indeed, as you have pointed out, the ASR system will mainly be utilized at the initial stage of robotic manipulation in many demonstration scenarios. The reason we provided a frequency statistic, where ASR might be utilized during every action update step, is primarily based on the following two considerations. **First**, as human-machine interaction scenarios become increasingly flexible and dynamic, some studies have emerged focusing on controlling robots through real-time voice commands to refine and adjust their movements. A similar concept is presented in “Yell At Your Robot” [1], where the ALOHA robot arm adapts to speech interventions during its movements, enabling it to better perform tasks. **Second**, It is foreseeable that human-machine interaction will become progressively complex in the future. Robots will also need to monitor speech information in real time during their movements, like humans. In particular, the speech signal will likely be processed alongside dynamic background sounds in the environment [2] to enhance the robot’s performance. For both of the above reasons, we provide a boundary statistic on the action frequency of robot with external speech processing modules, such as ASR.
> > > > >
> > > > > [1] Lucy Xiaoyang Shi, et.al. “Yell At Your Robot: Improving On-the-Fly from Language Corrections.” Robotics: Science and Systems, 2024.
> > > > >
> > > > > [2] Zeyi Liu, et.al. “ManiWAV: Learning Robot Manipulation from In-the-Wild Audio-Visual Data.” CoRL, 2024.

---

> > > > > > ### Comment · Reviewer_5GYV · 2024-11-29
> > > > > >
> > > > > > Thank you for your response. I agree that real-time performance is a critical issue, and your work shows potential for improving it compared to the cascading pipeline. However, it seems that this issue is not currently present in your experiments. Nonetheless, considering the potential of this work, I will maintain my score, although I believe it is borderline.

---

### Official Review · Reviewer_mePM · 2024-11-03

**Soundness:** 3
**Presentation:** 3
**Contribution:** 3
**Rating:** 6
**Confidence:** 4

**Summary:**

This paper creates a vision-language-action model by incorporating speech to create a VLAS model.  Two new datasets SQA and CSI are introduced.  A RAG setup is used for handling contextual queries.  Section 1 gives background for why they incorporated speech, including contextual queries.  The model is built on top of LLaVA.  A pretrained speech encoder was used.  The model is called, “Speech LLaVA.”  Comparable accuracy can be found on the CALVIN benchmark to traditional VLAs.  For customization tasks, this approach appears to do very well.  Section 2 describes related work.  Section 3 describes the approach, including the LLaVA-based approach, action tokenization / detokenization.  SQA dataset was created by sampling examples from the original LLaVA dataset and running them through a TTS system.  CSI dataset used 389 textual examples x 500 voices to result in 194k examples; training was interleavesd with text and audio.  A three stage process involving audio alignment, speech question answering and then robot manipulation fine tuning was performed.  Section 4 describes experiments and results across CALVIN, customized tasks and LLaVA.

**Strengths:**

This paper presents a natural extension to LLaVA to use speech input. With minor reservations about related work, the confounding nature of speech with and without context (and the lack of a text baseline), the exclusive use of TTS outputs and a few baselines, the paper is well-written and the results appear to be reproducible.

**Weaknesses:**

see above

**Questions:**

Some questions about the paper:

* The RAG pipeline appears to be intimately coupled with speech in this paper, but needn’t be.  Why wasn’t a baseline with language only inputs performed?  What would be the results?
* All the experiments appear to be with TTS output, which will certainly be biased.  Some results need to be with real speech.  Do you have any quantitative numbers with real speakers?  Even on a subset of any of the data you presented?
* The incorporation of RAG into the pipeline makes some parts of the paper confusing.  For example, in Section 4.1 it wasn’t clear if these leveraged the context part of the model (e.g., RAG) (e.g., Table 1).
* Why wasn’t OpenVLA shown in Table 1?


For related work, it would be good to cite at least the following:
* Other multi-modal models that incorporate audio should be included here.  Some missing references include, but there are likely more:
    * Unified-IO 2: https://arxiv.org/abs/2312.17172
    * Gemini (inherently multi-modal)
* Other VLMs, such as IDEFICS and Prismatic:
    * https://huggingface.co/blog/idefics
    * https://github.com/TRI-ML/prismatic-vlms

Other small things:
“pre-cess” -> “process” p2.
Inconsistency between VLAS and SpeechLLaVA terms.
“Absolute superiority” -> probably weaken this term.

---

> ### Author Response · Authors · 2024-11-24
> **Thanks for your affirmative review of this paper! We address your questions as follows.**
>
> **Weakness1**: `minor reservations about related work, the confounding nature of speech with and without context (and the lack of a text baseline), the exclusive use of TTS outputs and a few baselines`
>
> **A1**: We appreciate your valuable suggestions and are revising the paper to address these issues. We have enhanced the discussion of related works on multimodal LLMs that incorporate speech processing. Moreover, the experiments in which the RAG context is applied and those in which it is not are further clarified and explained. The text-only baselines used in the experiments have been further detailed and supplemented. In particular, we have conducted experiments using actual speech.
>
> ---
> **Q1**: `The RAG pipeline appears to be intimately coupled with speech in this paper, but needn’t be. Why wasn’t a baseline with language only inputs performed? What would be the results?`
>
> **A1**: As you pointed out, the RAG module can indeed be used independently of the speech input. Specifically, in the experiments conducted on the CALVIN benchmark, our model was evaluated using either text or speech instructions. In this part of the study, the RAG module is not utilized when the model processes speech input. Only for the personalization benchmark, the RAG module is used together with speech input.
>
> We have added new baselines using language-only inputs (speech input without voice RAG) on the personalization benchmark, as shown in the table. It is evident that removing the RAG module results in a significant decline in the performance of VLAS, highlighting the RAG’s effectiveness in acquiring and utilizing background knowledge.
> | Model | Ownership | Preference | Compound | Compound-Multistage (Stage-1) | Compound-Multistage(Stage-2) | Avg. |
> | --- | --- | --- | --- | --- | --- | --- |
> | VLAS            | 94.7% | 84.6% | 100%  | 100%  | 66.7% | 86.5% |
> | VLAS - RAG | 15.4% | 12.8% | 25.6% | 33.3% | 10.3% | 16.0% |
>
> **Q2**: `All the experiments appear to be with TTS output, which will certainly be biased. Some results need to be with real speech. Do you have any quantitative numbers with real speakers? Even on a subset of any of the data you presented?`
>
> **A2**: We have conducted new experiments to evaluate the performance of the proposed model when handling real speech instructions, both on the regular CALVIN benchmark and our personalization benchmark. These real speech commands were recorded from 10 persons. It can be found that our model continues to demonstrate good generalization to real audio. Given that our model is trained exclusively on a dataset synthesized from TTS, some performance degradation is acceptable when processing real audio.
> | Model | LH-1 | LH-2 | LH-3 | LH-4 | LH-5 | Avg. Len |
> | --- | --- | --- | --- | --- | --- | --- |
> | VLAS (TTS Speech) | 94.2% | 84.0% | 73.2% | 64.3% | 54.6% | 3.70 |
> | VLAS (Real Speech) | 93.6% | 82.8% | 71.6% | 61.4% | 51.3% | 3.61 |
>
> | Model | Ownership | Preference | Compound | Compound-Multistage (Stage-1) | Compound-Multistage(Stage-2) | Avg. |
> | --- | --- | --- | --- | --- | --- | --- |
> | VLAS (TTS Speech) | 94.7% | 84.6% | 100% | 100% | 66.7% | 86.5% |
> | VLAS (Real Speech) | 89.5% | 70% | 100% | 90.0% | 55.0% | 78.6% |
>
> **Q3**: `The incorporation of RAG into the pipeline makes some parts of the paper confusing. For example, in Section 4.1 it wasn’t clear if these leveraged the context part of the model (e.g., RAG) (e.g., Table 1).`
>
> **A3**: Sorry for your confusion! The experiments presented in Section 4.1 used only speech commands as input, without employing the voice RAG module. We are revising the description of the relevant sections to address the confusion you mentioned.
>
> **Q4**: `Why wasn’t OpenVLA shown in Table 1?`
>
> **A4**: We did not include OpenVLA as a baseline for comparison in Table 1 due to the following considerations:
>
> **a**. In terms of network structure, the VLA baseline used in our experiment is highly consistent with the OpenVLA model. The only difference is that OpenVLA employs two visual encoders (DinoV2 and SigLip) to extract distinct visual features, which are then concatenated as the final visual input. In contrast, the experimental VLA was implemented following RT-2, the work that first introduced the VLA concept, and therefore uses only a single visual encoder (CLIP). In other words, the experimental VLA baseline can be considered an approximate self-implementation of OpenVLA.
>
> **b**. In addition, OpenVLA utilizes the extensive open-source Open X-Embodiment dataset for pre-training, along with several additional training techniques to enhance the model’s performance. These training settings could impact the fairness of the comparative analysis between our model and the openVLA baseline.
>
> Thank you for pointing out some related work and other aspects we may have missed. We are revising the relevant content accordingly.

---

> > ### Comment · Reviewer_mePM · 2024-11-29
> >
> > Thank you for addressing some of my concerns, especially around adding a text baseline for RAG and adding a line for real speech to one of the tables.  Following up on the response:
> >
> > * UnifiedIO-2 - As it incorporates audio, it’s not clear that this couldn’t do speech recognition (e.g., in the VQA sense) as a task.  In fact, VQA is a task performed by UnifiedIO-2 (using text).  The statement around this is a bit strong in the related work.
> > * OpenVLA - Code and models have been released and could be used for common evaluation as at least a text-only baseline.  It's not clear why not just to use the model weights for evaluation.  Or possibly fine-tune as a comparison in Table 2.  Relatedly, it's not clear why roboflamingo was used other than the convenience of already working on the Calvin benchmark?  Were there other reasons?  Generally LLaVA-style models are simpler and just as (or more performant), and is consistent with the approach you use in this paper.
> > * Table 4 - The way I understand this table, the VLAS line only uses Whisper outputs.  I would like to at least see a sample of real speech here and a comparison of the other approaches on this subset of the dataset.  Ideally you'd have real speech outputs for all the text examples for evaluation.

---

> > > ### Author Response · Authors · 2024-12-01
> > > **Thank you for your helpful suggestions.**
> > >
> > > **Q1:**`UnifiedIO-2 - As it incorporates audio, it’s not clear that this couldn’t do speech recognition (e.g., in the VQA sense) as a task. In fact, VQA is a task performed by UnifiedIO-2 (using text). The statement around this is a bit strong in the related work.`
> > >
> > > **A1:** The primary focus of UnifiedIO-2 on the audio modality is processing natural sounds, such as event sounds, music tracks, and environmental sounds, rather than speech. According to their paper, they mainly evaluate their model on tasks such as text/image-to-audio generation, audio classification, audio captioning, and visual-audio fusion tasks. Considering that their architecture is applicable to speech data and can support speech tasks through fine-tuning, we will revise the statement to better clarify this possibility.
> > >
> > > **Q2:** OpenVLA - Code and models have been released and could be used for common evaluation as at least a text-only baseline. It's not clear why not just to use the model weights for evaluation. Or possibly fine-tune as a comparison in Table 2. Relatedly, it's not clear why roboflamingo was used other than the convenience of already working on the Calvin benchmark? Were there other reasons? Generally LLaVA-style models are simpler and just as (or more performant), and is consistent with the approach you use in this paper.
> > >
> > > **A2:** We did not use OpenVLA for evaluation mainly because, by the time they provided the corresponding inference interfaces, we had already implemented the VLA baseline described in the paper, which was based on the exact same architecture, except for one fewer visual encoder. Moreover, our VLA baseline supports input images from multiple views, while OpenVLA only supports the third-person view. Thus, our VLA baseline may achieve comparable or even better performance on the related benchmark. We employ Roboflamingo for comparison since the benchmark is proposed in this work, and it also achieves competitive performance.
> > >
> > > **Nevertheless, we are currently conducting experiments to evaluate OpenVLA on the same benchmark. We will share the results in a subsequent comment once they are available.**
> > >
> > > **Q3:** `Table 4 - The way I understand this table, the VLAS line only uses Whisper outputs. I would like to at least see a sample of real speech here and a comparison of the other approaches on this subset of the dataset. Ideally you'd have real speech outputs for all the text examples for evaluation.`
> > >
> > > **A3:** As for Table 4 in the paper, the LibriSpeech dataset is a real speech dataset recorded by various individuals for audiobooks. We only use Whisper for comparison here, as the Whisper (large v2) model used in our experiment was proven to be the most cutting-edge speech recognition model at the time. More detailed results can be seen in the Whisper[1] paper.
> > >
> > > [1] Radford, Alec, et al. “Robust speech recognition via large-scale weak supervision”, ICML, 2023.

---

> ### Author Response · Authors · 2024-12-02
> **Supplementary material for the experiment with OpenVLA.**
>
> We are evaluating both the pre-trained OpenVLA model and the fine-tuned OpenVLA model on the CALVIN benchmark. For the pre-trained OpenVLA model, we directly utilized the “openvla-7b-finetuned-libero-10” checkpoint, as it has been tailored for the Franka Emika Panda in a simulation environment, closely resembling our experimental setup. For the fine-tuned OpenVLA model, we followed their officially recommended fine-tuning setup. The CALVIN dataset was converted into the unified RLDS format to enable the fine-tuning of OpenVLA, as they suggested.
>
> Since running the full evaluation of above models on the CALVIN benchmark still requires an additional day, we present their performance on a subset of 30 long-horizon tasks before the end of the discussion. The performance comparison on this sub-dataset is sufficient to demonstrate that our own VLA baseline performs significantly better than the OpenVLA models. It can be observed from the table that:
>
> | Model (ABCD→D) | LH-1 | LH-2 | LH-3 | LH-4 | LH-5 | Avg. Len |
> | --- | --- | --- | --- | --- | --- | --- |
> | pre-trained OpenVLA | 0% | 0% | 0% | 0% | 0% | 0.000 |
> | fine-tuned OpenVLA | 16.7% | 0% | 0% | 0% | 0% | 0.167 |
> | VLA (our) | 90.0% | 86.7% | 66.7% | 63.3% | 56.7% | 3.633 |
>
> **a.** The pre-trained OpenVLA model fails to achieve zero-shot generalization on the CALVIN benchmark.
>
> **b.** Although the fine-tuned OpenVLA model can complete some tasks in the initial stage of the long-horizon sequence, its overall performance remains suboptimal. We hypothesize that the poor performance is due to its limitation of only supporting third-person view images. Whereas, the image input from the end-effector is crucial for the robot to effectively perform tasks such as picking and rotating. From some other studies[1][2]，we can also find that OpenVLA does not perform as expected. Another potential reason for this phenomenon is that OpenVLA is pre-trained in a large action space, encompassing various embodiments, which may differ significantly from the specific action space of the downstream task. This domain gap may also negatively affect its performance[3]. Therefore, our own VLA model is essential and provides a more reliable baseline for the experiments.
>
>
> **Ref:**
>
> [1] Li, Qixiu, et al. "CogACT: A Foundational Vision-Language-Action Model for Synergizing Cognition and Action in Robotic Manipulation." arXiv:2411.19650, 2024.
>
> [2] Liu, Songming, et al. "RDT-1B: a Diffusion Foundation Model for Bimanual Manipulation." arXiv:2410.07864, 2024.
>
> [3] Szot, Andrew, et al. "Grounding Multimodal Large Language Models in Actions." NeurIPS, 2024.

---

### Official Review · Reviewer_TJDB · 2024-11-04

**Soundness:** 2
**Presentation:** 4
**Contribution:** 1
**Rating:** 5
**Confidence:** 5

**Summary:**

The paper introduces VLAS, a novel vision-language-action model designed to facilitate robot manipulation tasks by integrating speech as a primary instruction modality, bypassing traditional speech-to-text conversion methods. This approach enables VLAS to capture essential non-verbal cues, such as voiceprints, which assist in executing personalized tasks based on individual user needs. Built on the LLaVA model, VLAS employs three training stages to process multimodal inputs across text, image, and speech, aligning these representations for seamless action generation. The authors also developed two datasets, SQA and CSI, to enhance the model’s handling of speech instructions, alongside a voice retrieval-augmented generation (RAG) system to access individualized knowledge. Experimental results demonstrate VLAS’s superior performance on complex, customized robot tasks, particularly when handling personalized commands, outperforming other models that rely on separate speech recognition systems.

**Strengths:**

1. The paper introduces VLAS, a vision-language-action model evaluated on a comprehensive set of tasks, including customized tasks that leverage speech input for personalized robot manipulation.

2. Additionally, the authors contribute two new datasets, SQA and CSI, which enhance the model's ability to process multimodal inputs by including speech instructions alongside visual data.

**Weaknesses:**

1. The necessity of incorporating raw speech, given the availability of video and language (transcripts), remains unclear. Many tasks focus on ownership, and contemporary speech recognition models are capable of discerning individual identities through voice characteristics. Integrating recognized user identity and vocal nuances as metadata in text input might achieve similar effects in current vision-language models (VLMs).

2. While speech transcripts alone may lack certain nuances, the video input could compensate for these through visual cues like gestures. The paper’s limited treatment of emotion in its framework is another concern, especially as language models like ChatGPT already offer emotion detection from text ("Emotion detection GPT"), which could outperform humans [8].

3. Overall, the additional complexity introduced by adding raw speech input to video and transcript data may not significantly enhance model performance in a meaningful way.

4. The paper might benefit from using more recent multimodal VLMs, such as MA-LMM [1], Chat-UniVi [2], ST-LLM [3], LLaVA-NeXT-Video [4], VideoChat2 [5], VideoLLaMA 2 [6], and Video-LLaVA [7], to contextualize the VLAS model's contributions within the latest advancements.

References:

[1] He, Bo, et al. "Ma-lmm: Memory-augmented large multimodal model for long-term video understanding." Proceedings of the IEEE/CVF Conference on Computer Vision and Pattern Recognition. 2024.

[2] Jin, Peng, et al. "Chat-univi: Unified visual representation empowers large language models with image and video understanding." Proceedings of the IEEE/CVF Conference on Computer Vision and Pattern Recognition. 2024.

[3] Liu, Ruyang, et al. "St-llm: Large language models are effective temporal learners." European Conference on Computer Vision. Springer, Cham, 2025.

[4] https://huggingface.co/llava-hf/LLaVA-NeXT-Video-7B-hf

[5] Li, KunChang, et al. "Videochat: Chat-centric video understanding." arXiv preprint arXiv:2305.06355 (2023).

[6] Cheng, Zesen, et al. "VideoLLaMA 2: Advancing Spatial-Temporal Modeling and Audio Understanding in Video-LLMs." arXiv preprint arXiv:2406.07476 (2024).

[7] Lin, Bin, et al. "Video-llava: Learning united visual representation by alignment before projection." arXiv preprint arXiv:2311.10122 (2023).

[8] Elyoseph, Zohar, et al. "ChatGPT outperforms humans in emotional awareness evaluations." Frontiers in Psychology 14 (2023): 1199058.

**Questions:**

"The voice retrieval-augmented generation (RAG) (Zhao et al., 2024) is subsequently proposed":

If this method was previously developed by Zhao et al. (2024), why is it described as "subsequently proposed" here? Is the paper introducing a new variant or application of this technique, or is it simply implementing the existing RAG approach as described in Zhao et al. (2024)?

---

> ### Author Response · Authors · 2024-11-24
> **Thanks for your detailed review of this paper! We will explain your concerns point by point.**
>
> **Weakness1**: `The necessity of incorporating raw speech, given the availability of video and language (transcripts), remains unclear. Many tasks focus on ownership, and contemporary speech recognition models are capable of discerning individual identities through voice characteristics. Integrating recognized user identity and vocal nuances as metadata in text input might achieve similar effects in current vision-language models (VLMs).`
>
> **A1**: While the majority of current multi-modal LLMs or VLA models demonstrate effective performance in scenarios involving images/videos and textual input, we argue that the integration of raw speech modalities remains valuable for several reasons.
>
> **a**. Speech represents a unique modality of information, fundamentally distinct from text, as it conveys richer non-semantic content. This non-semantic information, including speaker identity, emotion, and intonation, can enhance the capabilities of VLA and MLLM models, enabling them to better understand and perform assigned tasks. An increasing number of studies have begun to focus on integrating raw speech into Multimodal LLMs, such as GPT-4o [1] and VITA [2].
>
> **b**. Though it is possible to achieve similar results by employing an additional speech processing module to recognize speech content while simultaneously summarizing vocal nuances as meta-text, this approach presents several challenges. **First**, the modular pipeline increases the system’s complexity and negatively impacts the robot’s execution efficiency. As shown below, the robot policy employing the ASR+VLA approach consistently lags behind the proposed VLAS model in terms of inference speed during the experiments. **Second**, in the cascaded pipeline, the pre-trained ASR is designed as a generalist for diverse speech inputs, which limits its sensitivity to robot manipulation commands. As a result, this approach can introduce cumulative errors that significantly harm the model performance. This issue can also be observed in the table and the paper.
> | Model | Actions / Sec (Hz) | Avg. Len |
> | --- | --- | --- |
> | VLA (r=1) | 1.89  | 2.30 |
> | VLAS (r=1) |  1.17 | 2.02 |
> | VLA (r=5) | 3.60 | 3.80 |
> | ASR + VLA (r=5) | 0.17 | 3.13 |
> | VLAS (r=5) | 2.50 | 3.70 |
>
> In conclusion, while modular solutions are effective, developing a unified end-to-end robot policy model for speech processing remains highly valuable. It may offer an alternative approach distinct from mainstream methods, fostering further scientific exploration.
>
> **Reference**
>
> [1] https://openai.com/index/hello-gpt-4o/
>
> [2] Fu, Chaoyou, et al. "VITA: Towards Open-Source Interactive Omni Multimodal LLM." arXiv preprint arXiv:2408.05211 (2024).
>
> **Weakness2**: `While speech transcripts alone may lack certain nuances, the video input could compensate for these through visual cues like gestures. The paper’s limited treatment of emotion in its framework is another concern, especially as language models like ChatGPT already offer emotion detection from text ("Emotion detection GPT"), which could outperform humans.`
>
> **A2**: As you mentioned, incorporating visual cues such as gestures is a potentially feasible method. However, this approach is neither the most direct nor the most convenient solution, especially for robot manipulation tasks.
>
> **a**. In current robotic manipulation tasks, vision sensors are typically positioned to monitor the operation space, either mounted on the robotic arm’s end-effector or placed at a third-person viewpoint focusing on the robotic arm. The person issuing commands is usually outside the visual sensory space, partly to ensure human safety. Therefore, compensating for certain nuances through visual information necessitates additional devices and algorithms to track individual behavior, which inevitably increases the system’s cost and complexity.
>
> **b**. The VLA model is designed to output Cartesian coordinate actions for the robot to execute, placing greater emphasis on fine-grained location information of objects in the currently observed image. Therefore, though many advanced video-text LLMs exist, in the field of VLA, exemplified by RT-2 and OpenVLA, both utilize straightforward image-text LLMs as the foundation to simultaneously ensure the robot’s performance and efficiency. Capturing human intentions through video can be challenging in terms of both performance and speed.
>
> **c**. Although this paper does not address emotional features, the proposed paradigm is equally applicable to them. All that is required is to integrate a pre-trained emotion recognition module and a RAG module for emotion processing. We chose voiceprint as the focus of our research because it involves a broader range of scenarios compared to emotions. Since we focus on manipulation tasks rather than conversational tasks like ChatGPT, the impact of emotions on robot manipulation is relatively limited.

---

> > ### Author Response · Authors · 2024-11-24
> > **Response cont.**
> >
> > **Weakness3**: `Overall, the additional complexity introduced by adding raw speech input to video and transcript data may not significantly enhance model performance in a meaningful way.`
> >
> > **A3**: Please allow us to clarify that the purpose of this work is not to improve the performance of the VLA on standard benchmarks or to optimize the VLM foundation. Instead, our goal is to propose a new robot manipulation paradigm that:
> >
> > **a**. uses a unified model to process information from multiple modalities (image, text, and speech), without compromising the performance of the original VLA.
> >
> > **b**. mitigates performance decay caused by cumulative errors in the cascaded pipeline.
> >
> > **c**. is better suited for certain scenarios requiring personalized background knowledge.
> >
> > **Weakness4**: `The paper might benefit from using more recent multimodal VLMs to contextualize the VLAS model's contributions within the latest advancements.`
> >
> > **A4**: Thanks for your helpful suggestions. Utilizing the latest VLM model as the foundation can absolutely enhance the effectiveness of our approach. The approach presented in this paper is applicable to other VLM models as well. We chose LLaVA v1.5 as the foundation mainly due to its significant influence, widespread popularity, and full accessibility. We will synchronize and update the foundation model to the latest version in our future work.
> >
> > ---
> > **Q1**: `"The voice retrieval-augmented generation (RAG) (Zhao et al., 2024) is subsequently proposed": If this method was previously developed by Zhao et al. (2024), why is it described as "subsequently proposed" here? Is the paper introducing a new variant or application of this technique, or is it simply implementing the existing RAG approach as described in Zhao et al. (2024)?`
> >
> > **A1**: Sorry for making you confused. The paper implements a simple RAG module similar to existing methods. However, while most RAGs utilize text or images to retrieve relevant knowledge, we leverage voiceprints instead.

---

> > > ### Comment · Reviewer_TJDB · 2024-11-28
> > > **Thank You for Your Response — Remaining Issues and Concerns**
> > >
> > > Thanks authors for the response and additional evidence. My responses to your rebuttal are outlined below:
> > >
> > > **RE A1.a:**
> > >
> > > *"Speech represents a unique modality of information, fundamentally distinct from text, as it conveys richer non-semantic content, including speaker identity, emotion, and intonation…”*
> > >
> > > Unfortunately, the experiments in the paper do not convincingly demonstrate the unique benefits of using raw speech over speech-converted transcripts. As noted in my review, the current experiments focus primarily on challenges related to user identity, which can be effectively addressed using existing speech recognition methods. Similarly, for emotion, text-based emotion recognition has shown strong results, as demonstrated in [1]: *GPT-4V with emotion: A zero-shot benchmark for Generalized Emotion Recognition*.
> > >
> > > To provide compelling evidence, it would be helpful to compare against baselines that incorporate cutting-edge speech recognition and emotion recognition methods. For instance, user identity from speech recognition or emotion metadata from text-based analysis could be appended to each transcript and tested for performance improvements.
> > >
> > > [1] Lian, Zheng, Licai Sun, Haiyang Sun, Kang Chen, Zhuofan Wen, Hao Gu, Bin Liu, and Jianhua Tao. "Gpt-4v with emotion: A zero-shot benchmark for generalized emotion recognition." Information Fusion 108 (2024): 102367.
> > >
> > > **RE A1.b:**
> > >
> > > *"Second, in the cascaded pipeline, the pre-trained ASR is designed as a generalist for diverse speech inputs, which limits its sensitivity to robot manipulation commands. As a result, this approach can introduce cumulative errors that significantly harm the model performance."*
> > >
> > > The claim that ASR is a generalist and limited for robot manipulation commands may not fully align with real-world usage. In most real-world scenarios, especially in household or everyday environments, humans typically communicate with robots using natural language commands including general elements, like "please pick up the apple". These commands fall well within the capabilities of modern ASR systems, which are adept at processing straightforward and commonly used language. While domain-specific jargon may occasionally be used by robotics experts, the overarching goal of AI and robotics research is to design systems that serve the general public (individuals who are not experts and rely on intuitive, accessible interactions).
> > >
> > > In addition, the robustness of LLMs or VLMs (e.g., ChatGPT) to handle noisy text and infer meaning from context suggests that cumulative errors in an ASR + VLA pipeline might have minimal impact on performance. These models often use context to resolve ambiguities and tolerate inaccuracies in input, making them suitable for natural language tasks.
> > >
> > > *"First, the modular pipeline increases the system’s complexity and negatively impacts the robot’s execution efficiency. As shown below, the robot policy employing the ASR+VLA approach consistently lags behind the proposed VLAS model in terms of inference speed."*
> > >
> > > While modular pipelines may increase processing time, this impact can be mitigated by leveraging cloud-based ASR solutions, such as the OpenAI Whisper API, which provide significantly faster processing speeds. With reliable internet connections being commonplace in most households, network latency should not be a significant issue. Additionally, most household tasks are not highly time-sensitive, making a short delay, e.g. 1-2 seconds, generally acceptable for practical use. To strengthen their argument, the authors could focus their experiments on time-sensitive tasks where such delays would critically impact performance or user experience.
> > >
> > > **RE A2:**
> > >
> > > *"Therefore, compensating for certain nuances through visual information necessitates additional devices and algorithms to track individual behavior, which inevitably increases the system’s cost and complexity."*
> > >
> > > It is true that incorporating additional systems to observe users could increase costs and complexity. However, this trade-off should be assessed in terms of the potential benefits. Research by Albert Mehrabian indicates that communication is 55% nonverbal, 38% vocal (intonation, tone, etc.), and only 7% words [2]. This highlights the crucial role of non-verbal cues -- such as eye gaze, body language, and facial expressions -- in effective communication. The visual observation of users also naturally provides robots with valuable insights into **emotions**, an aspect the authors mention in the paper but do not sufficiently evaluate. Relying solely on vocal input without integrating human observation risks missing the majority of communicative information. Investigating whether visual cues in videos can already offer complementary or even superior information compared to speech is essential to better understand the added value of incorporating speech.
> > >
> > > [2] Mehrabian, Albert. "Communication without words." In Communication theory, pp. 193-200. Routledge, 2017.

---

> ### Author Response · Authors · 2024-12-01
> **Thank you for your detailed response.**
>
> Thank you for your detailed response. We would like to address your remaining concerns point by point.
>
> **To RE 1:**
>
> `Unfortunately, the experiments in the paper do not convincingly demonstrate the unique benefits of using raw speech over speech-converted transcripts. … Similarly, for emotion, text-based emotion recognition has shown strong results`
>
> **First**, we would like to claim that our framework is adaptable for emotion processing. The extraction of speaker and emotion embeddings follows a similar approach, where spectral extraction is applied, followed by a network that compresses it into hidden representations. Then, these representations are used with their corresponding task heads. We use identity information because it is the most suitable for the robot manipulation tasks and covers a wide range of applications. Although emotion, intonation, and other factors may be useful for the VLM, their impact on the VLA task is currently limited.
>
> **Second**, our paper introduces an end-to-end robot policy that maps raw speech directly to robot actions. This has not been explored in existing research. Meanwhile, the unified architecture helps mitigate cumulative errors, as demonstrated in Section 4.1.
>
> **Last but most importantly**, when raw speech is used as input, the model offers a potential way for non-semantic content (such as identity) to directly influence actions, which is both intuitive and akin to human interactions. Given the current lack of diverse (speech, action) data, we propose Voice RAG as an alternative solution. However, with sufficient (speech, action) data, our model can inherently learn this personalization, even without the RAG module. In this case, the cascading approach requires designing multiple models, each dedicated to addressing a specific aspect of the original speech. This makes the system more complex and diminishes its extensibility.
>
> `To provide compelling evidence, it would be helpful to compare against baselines that incorporate cutting-edge speech recognition and emotion recognition methods.  For instance, user identity from speech recognition or emotion metadata from text-based analysis could be appended to each transcript and tested for performance improvements.`
>
> In fact, similar experiments have already been presented in our paper. As demonstrated in Section 4.2, we implement a VLA + RAG baseline for comparison with our VLAS. The VLA receives the textual transcript, augmented with near-ground-truth identities from the speaker identification model. The results show that this cascading method does not demonstrate superiority, lagging behind our model by a margin of 4.5%. Moreover, on the standard CALVIN benchmark, where speech is used without identity, our VLAS outperforms the VLA + ASR (Whisper), achieving an average length of 3.70 compared to 3.13.
>
> **To RE 2:**
>
> `The claim that ASR is a generalist and limited for robot manipulation commands may not fully align with real-world usage. … While domain-specific jargon may occasionally be used by robotics experts, the overarching goal of AI and robotics research is to design systems that serve the general public.`
>
> Sorry for the misunderstanding. What we intend to convey here is not that general ASR is ineffective. Rather, we believe the model could perform better if it were to incorporate domain-specific knowledge. It's kind of like training people to specialize in a particular task. As demonstrated in Section 4.4, our VLAS-base also works as a general speech recognition model and achieves performance comparable to the Whisper model (2.7% vs 2.79%). However, when used for robot manipulation with domain specific data, our model further help mitigate the cumulative error.
>
> `In addition, the robustness of LLMs or VLMs (e.g., ChatGPT) to handle noisy text and infer meaning from context suggests that cumulative errors in an ASR + VLA pipeline might have minimal impact on performance. These models often use context to resolve ambiguities and tolerate inaccuracies in input, making them suitable for natural language tasks.`
>
> Notably, although VLA is fine-tuned from VLM, it does not perform as robustly as VLM. Roboflamingo [1] conducts an experiment in which the language instruction for VLA is enriched(using different transcripts with similar meanings). The results show a significant decline in performance. Therefore, the subtle recognition errors from ASR may be amplified by the VLA model, resulting in significant cumulative errors, as demonstrated in Section 4.1.
>
> [1] Li, Xinghang et al. “Vision-Language Foundation Models as Effective Robot Imitators”, ICLR, 2024.

---

> ### Author Response · Authors · 2024-12-01
> **Response cont.**
>
> `While modular pipelines may increase processing time, this impact can be mitigated by leveraging cloud-based ASR solutions, such as the OpenAI Whisper API.`
>
> We agree that cloud-based ASR solutions can be effectively utilized. However, they also come with their own set of limitations, such as the high cost, network transmission delays, security and privacy risks. In such cases, our method offers an alternative, allowing users to make choices based on their specific needs. Particularly in the domains of security and privacy, transferring speech data to the cloud is a highly sensitive activity. This is likely to face potential risks of counterfeiting and deception. Thus, many cloud service companies are extremely cautious when it comes to handling speech data.
>
> **To RE 3:**
>
> `Research by Albert Mehrabian indicates that communication is 55% nonverbal, 38% vocal   (intonation, tone, etc.), and only 7% words.  This highlights the crucial role of non-verbal cues -- such as eye gaze, body language, and facial expressions -- in effective communication. …. Relying solely on vocal input without integrating human observation risks missing the majority of communicative information. Investigating whether visual cues in videos can already offer complementary or even superior information compared to speech is essential to better understand the added value of incorporating speech.`
>
> **First**, please allow us to state once again that most VLA models currently do not use video as input to understand the scene. This is because video processing may impose a substantial computational burden on robot inference, surpassing the performance improvements it offers. A standard VLA model provides an action frequency of 2-4Hz. However, if the video modality with multiple frames is used, the processing speed could decrease significantly, making it unsuitable for robot manipulation. This is why current state-of-the-art VLAs [1][2][3][4][5] rely on image inputs, and the largest public datasets [5] provide only images either from the end-effector (focusing on the objects) or from a third-person view (focusing on the robot arm, not the person).
>
> **Second**, as you pointed out, according to Albert Mehrabian, communication is 55% nonverbal, 38% vocal (intonation, tone, etc.), and only 7% words. In current studies on robot policies, visual information (accounting for part of the 55%) and words (7%) have been widely utilized[1][2][3][4][5]. However, efforts to utilize vocal information remain very scarce (perhaps 0%), especially the vocal information account for 38% of the communication. Therefore, we believe that it is imperative to explore the potential use of vocal information.
>
> **Therefore**, introducing identity information from speech into current VLAs and combining it with the image modality may be a better approach than visual-only methods.
>
> [1] Li, Xinghang et al. “Vision-Language Foundation Models as Effective Robot Imitators”, ICLR, 2024.
>
> [2] Brohan, Anthony, et al. “RT-2: Vision-Language-Action Models Transfer Web Knowledge to Robotic Control”, CoRL, 2023.
>
> [3] Belkhale, Suneel, et al. “RT-H: Action Hierarchies Using Language”, RSS, 2024.
>
> [4] Kim, Moo Jin, et al. “OpenVLA: An Open-Source Vision-Language-Action Model”, arXiv:2406.09246, 2024.
>
> [5] Padalkar, Abhishek, et al. “Open X-Embodiment: Robotic Learning Datasets and RT-X Models”, arXiv:2310.08864, 2023.

---

> > ### Comment · Reviewer_TJDB · 2024-12-03
> > **Thank you**
> >
> > Thank you for your response. I appreciate the clarifications provided. While I’m willing to raise my score to a weak reject, I still have some reservations. Specifically, I believe additional experiments are needed to explore factors beyond identity, such as vocal characteristics (intonation, tone, etc.) and emotional cues, to better demonstrate the benefits of directly using speeches rather than speech-converted transcripts. Additionally, a 4.5% improvement, while measurable, does not strike me as sufficiently significant to justify the claims made.

---

### Official Review · Reviewer_PCso · 2024-11-04

**Soundness:** 3
**Presentation:** 3
**Contribution:** 2
**Rating:** 8
**Confidence:** 4

**Summary:**

This paper recognizes the problems of losing auxiliary semantics and complicating the system in performing transcription on human speech as robot manipulation command. It proposes to include the speech modality into an end-to-end LVA model, resulting in the VLAS model.

The VLAS model directly process textual and speech instructions with vision and proprioception, and generates the robot action in an auto-regressive way. A VQA dataset and a he CALVIN dataset are augmented with the speech instructions to train the VLAS model, and the latter one is further used to tune the model on robotic manipulation tasks.

The experiments demonstrate the promising results of VLAS, and showcases interesting way of commanding robots through the ownership, preference, and compound tasks.

**Strengths:**

1. The paper is well-written and easy to read. It provides efficient background information and detailed illustration of the proposed method.
2. The speech-driven manipulation task is novel. The design of storing user information within the dataset and generate the actions with RAG is smart, demonstrating interesting use cases concerning user ownership and preference.
3. The real-world demonstration further showcase the capability of the model.

**Weaknesses:**

1. Despite the common issue of inference speed in tuning a large VL model as a VLA model, the authors should still provide an analysis on the time required on using the model, especially the bottleneck (is it the audio processing part?) as well as the possible strategies to boost it.
2. The evaluation on robotic manipulation is not efficient. It should be evaluated whether the policy generalizable to novel scenes (like training on CALVIN ABC and test on D, or in other simulations) or tasks. More importantly, if the model requires further training on novel cases, whether the speech processing modules can be transferred.

**Questions:**

1. I wonder if there are additional use case that leverage more semantics information from the speech, like the emotion?
2. How long historical information can the model handle? If there are multiple users in history and their ownership violates each other, will the model be able to handle this?

---

> ### Author Response · Authors · 2024-11-24
> **Thanks for your detailed review of this paper! We hereby address your concerns as follows.**
>
> **Weakness1**: `Despite the common issue of inference speed in tuning a large VL model as a VLA model, the authors should still provide an analysis on the time required on using the model, especially the bottleneck (is it the audio processing part?) as well as the possible strategies to boost it.`
>
> **A1**: As you pointed out, analyzing inference efficiency is crucial when fine-tuning a VLM into a VLA model for robotic control. Indeed, this paper employs two key optimizations to enhance the inference speed of the models: downsampling the speech spectrogram and implementing an action update strategy with multi-step prediction and execution.
>
> **a**. Speech spectrogram downsampling is a widely used strategy to accelerate speech signal processing, where adjacent x-frame spectrograms are aggregated into a single-frame feature through a reshaping operation, effectively reducing the time dimension length. In our experiments, we used the x = 5. Since the effectiveness of this approach has been validated in numerous speech recognition and generation tasks, we did not perform additional related experiments.
>
> **b**. Given that the state of the environment typically does not change significantly over a short period, our work adopts a simple yet effective multi-step prediction and execution policy. Specifically, we set the number of steps for both VLA and VLAS to r=5. As shown in the following table, when  r=5 , both the VLA and VLAS models achieve significant speedups while also demonstrating improved performance on the CALVIN benchmark. The frequency is measured using a single NVIDIA A800 GPU.
> | Model | Actions / Sec (Hz) | Avg. Len |
> | --- | --- | --- |
> | VLA (r=1) | 1.89  | 2.30 |
> | VLAS (r=1) |  1.17 | 2.02 |
> | VLA (r=5) | 3.60 | 3.80 |
> | ASR + VLA (r=5) | 0.17 | 3.13 |
> | VLAS (r=5) | 2.50 | 3.70 |
>
> We also supplemented the results with an analysis of inference speed and performance for different values of  r. The table indicates that r=5 achieves an optimal balance between inference efficiency and manipulation performance.
> | Model | Actions / Sec (Hz) | Avg. Len |
> | --- | --- | --- |
> | VLAS (r=1) | 1.17 | 2.02 |
> | VLAS (r=5) | 2.50 | 3.70 |
> | VLAS (r=12) | 2.88 | 3.35 |
> | VLAS (r=20) | 3.80 | 0.70 |
>
> In conclusion, although the introduction of the speech modality reduces the model’s inference speed, we have mitigated this issue using the aforementioned techniques. Other potential solutions to enhance inference efficiency include techniques such as model quantization and speculative sampling. Our future work will involve the development of a Mamba-based VLAS model to further enhance inference speed.
>
> **Weakness2**: `The evaluation on robotic manipulation is not efficient. It should be evaluated whether the policy generalizable to novel scenes (like training on CALVIN ABC and test on D, or in other simulations) or tasks.`
>
> **A2**: To better evaluate our model’s generalization capability to novel scenes, we conducted experiments where the model was trained on dataset ABC and tested on dataset D. It can be observed that, although all the models experienced performance degradation due to the domain gap, our VLAS model achieved performance comparable to RoboFlamingo and VLA on the CALVIN benchmark, while outperforming the other models.
> | Model (ABC->D) | LH-1 | LH-2 | LH-3 | LH-4 | LH-5 | Avg. Len |
> | --- | --- | --- | --- | --- | --- | --- |
> | MCIL | 0.304 | 0.013 | 0.002 | 0.000 | 0.000 | 0.31 |
> | HULC | 0.418 | 0.165 | 0.057 | 0.019 | 0.011 | 0.67 |
> | RT-1 | 0.533 | 0.222 | 0.094 | 0.038 | 0.013 | 0.90 |
> | VLA | 0.831 | 0.584 | 0.347 | 0.231 | 0.151 | 2.14 |
> | RoboFlamingo | 0.824 | 0.619 | 0.466 | 0.331 | 0.235 | 2.48 |
> | VLAS (Text) | 0.859 | 0.592 | 0.385 | 0.259 | 0.176 | 2.27 |
> | ASR + VLA (Speech) |0.747 | 0.541 | 0.384 | 0.241 | 0.165 | 2.04 |
> | VLAS (Speech) | 0.872 | 0.642 | 0.409 | 0.281 | 0.196 | 2.40 |
>
> Moreover, we conducted similar experiments on our personalization benchmark. The results demonstrate that our model is capable of handling novel scenes.
> | Model (ABC->D)| Ownership | Preference | Compound | Compound-Multistage (Stage-1) | Compound-Multistage(Stage-2) | Avg. |
> | --- | --- | --- | --- | --- | --- | --- |
> | VLA | 20.5% | 5.1% | 0.0% | 10.3% | 0.0% | 6.4% |
> | VLAS | 64.1% | 61.5% | 87.2% | 74.4% | 7.7% | 55.1% |
> | VLA + RAG | 82.1% | 71.8% | 84.6% | 82.1% | 10.3% | 62.2% |
> | VLAS - RAG | 15.4% | 23.1% | 0.0% | 12.8% | 0.0% | 9.6% |

---

> ### Author Response · Authors · 2024-11-24
> **Response cont.**
>
> ---
> **Q1**: `I wonder if there are additional use case that leverage more semantics information from the speech, like the emotion?`
>
> **A1**: Though our paper focuses on the identity information in speech, the overall pipeline can be readily adapted to tasks involving emotions. All we need to do is integrate a pre-trained emotion recognition module along with an RAG module for emotion handling. We focus on voiceprints in our research primarily because they cover a broader range of scenarios compared to emotions in the context of robotic manipulation.
>
> Some potential use cases for emotion: For example, bringing your favorite food to you when you’re in a bad mood. Or perhaps, when responding to positive emotions, robots might perform tasks at a slower pace but with greater precision. On the other hand, when reacting to negative emotions, robots might prioritize speed, completing tasks more quickly at the expense of some accuracy.
>
> **Q2**: `How long historical information can the model handle? If there are multiple users in history and their ownership violates each other, will the model be able to handle this?`
>
> **A2**: For a single round of dialogue, our model can accept an input length of up to 2048 tokens, matching that of the LLaVA v1.5 model. Currently, our model supports processing only one user’s speech command per round. However, for the robot manipulation task, we treat it as a multi-round dialogue in which the observed images and the speech instruction are updated over time steps. When instructions conflict with each other, the robot will always prioritize the latest user and its preferences. We think this is a very good question. In the future, we may address such situations more flexibly by incorporating different priority levels into the RAG module.

---

> ### Author Response · Authors · 2024-12-02
> **Friendly Reminder Regarding Upcoming Rebuttal Deadline**
>
> Dear Reviewer PCso:
>
> I hope this message finds you well. This is a gentle reminder regarding the review of our manuscript. We deeply appreciate the invaluable comments and feedback provided by reviewers. They are instrumental in enhancing the quality of our research. As per the schedule, the rebuttal phase is drawing to a close. We understand that you have a demanding schedule and a multitude of responsibilities, but we are keen to receive your feedback before the deadline. This will afford us the opportunity to address any questions or concerns you may have raised in a timely manner. We are eager to incorporate your insights to refine our work and would be grateful if you could share your thoughts prior to the rebuttal deadline.
>
> Thank you very much for your hard work and support. Your dedication to the review process is greatly appreciated.

---

> ### Comment · Reviewer_PCso · 2024-12-03
> **Response to the rebuttal**
>
> Thanks to all the authors for their feedback, and I decide to raise my grading. Looking forward to GPT-like interaction with robots via voice commands :D

---

> > ### Author Response · Authors · 2024-12-03
> >
> > Thank you very much for your time to reviewing our paper and providing such helpful feedback. We sincerely appreciate your recognition of our work. We will continue striving to enhace the integration of speech into robotics applications.

---

### Author Response · Authors · 2024-12-04
**Overall response**

We sincerely thank all the reviewers for their reviews and suggestions! We have made every effort to address the mentioned typos, clarifications, and concerns. With the discussion phase drawing to a close, the major improvements have been summarized as follows:

1. A more detailed inference efficiency analysis of VLAS has been provided to illustrate how the potential inference speed challenges are addressed when integrating the speech modality into the vision-language-action (VLA) model.
2. We have thoroughly evaluated our model’s ability to generalize to novel scenes, for example, training on CALVIN ABC and testing on D.
3. Our proposed Voice RAG module is employed either in a coupled or decoupled manner with the speech input to facilitate more comprehensive ablation studies.
4. In addition to TTS-generated audio, we have incorporated real speech instructions to further validate the effectiveness of our model.
5. We have presented some real-world experimental results and a comparison between openVLA and our self-implemented VLA baseline, to address the reviewers’ concerns.

---

### Meta-Review · Area_Chair_f212 · 2024-12-23

**Metareview:**

The paper proposes VLAS, a vision-language-action (VLA) model built upon LLaVA for commanding robot manipulation using natural language speech as the primary input modality. This is in contrast to standard approaches that first convert speech to text using an off-the-shelf speech recognizer. Reasoning directly over speech affords the ability to reason over aspects of the commands that are not natively captured by speech recognizers (e.g., voiceprints, which can be important to understanding personalized commands). VLAS employs a three stage training process that includes audio alignment, speech question answering, and action fine-tuning. In addition to VLAS, the paper introduces two new datasets to facilitate training the model to reason over text, images, speech, and actions. Additionally, they introduce a retrieval-augmented generation (RAG) framework to support the model's ability to handle tasks that require reasoning over individual-specific knowledge. The paper evaluates VLAS on a series of custom robot tasks, with results showing that VLAS outperforms existing methods, including those that rely on separate speech recognition tools, particularly in the context of handling personalized tasks.

The paper was reviewed by four referees, and the evaluation involved a significant amount of discussion between the authors, reviewers, and later the AC. All four reviewers emphasized that the paper is well written and easy to read. There was some disagreement regarding the importance of having a model capable of reasoning directly over raw speech as opposed to using off-the-shelf speech processing models. There was a healthy discussion regarding this point between the authors and Reviewer TJDB, both of whom make valid points. As someone who has worked on language understanding in the context of robots for more than a decade, the AC understands the potential benefits of being able to handle raw speech as the primary input modality. Reviewer TJDB's points that some of the information that is conveyed via speech but not captured by a TTS model, could be added as an auxiliary input alongside the recognized text may be reasonable. However, it is not clear that such a two-stage approach would be appropriate in all settings or equally as performant. Of course, more extensive comparisons to such two-stage approaches would go a long way in supporting the importance of reasoning over raw speech (the comparison to the VLA+RAG baseline helps, but there were concerns about the significance of the performance gap and whether this could be attributed to parameter tuning). Other concerns raised by the reviewers include the exclusive use of TTS-based synthetic speech for the initial evaluation and whether the results would generalize to real speech, the lack of a sufficient evaluation of inference speed/efficiency, the coupling of the RAG pipeline with speech, and the absence of relevant baselines (namely, OpenVLA and a text-based baseline). The authors clearly put considerable effort into trying to address these concerns during the rebuttal period, which included performing new experiments involving VLAS' inference efficiency, the model's generalizability given novel scenes, it's performance given real speech, real-world results, and a comparison to OpenVLA.

The AC believes that the paper very much benefited from the discussion period. Updating the paper to incorporate this discussion as well as the additional experiments would go a long way in clarifying the significance of its contributions.


Note: Reviewer PCso is very new to research and has the least experience among the four reviewers. The AC took this into account when making their recommendation.

**Additional Comments On Reviewer Discussion:**

There was a healthy amount of discussion between the reviewers and the authors. The AC reached out to the reviewers at various points during the process to encourage them to read and respond to the authors' rebuttal as well as to try and help them come to a consensus on the paper.

---

### Decision · Program_Chairs · 2025-01-22

Accept (Poster)